# *Cis*-regulatory interfaces reveal the molecular mechanisms underlying the notochord gene regulatory network of *Ciona*

Lenny J. Negrón-Piñeiro[1,4], Yushi Wu[1,4], Sydney Popsuj[2], Diana S. José-Edwards [3], Alberto Stolfi[2] & Anna Di Gregorio [1] ✉

Tissue-specific gene expression is fundamental in development and evolution, and is mediated by transcription factors and by the *cis*-regulatory regions (enhancers) that they control. Transcription factors and their respective tissue-specific enhancers are essential components of gene regulatory networks responsible for the development of tissues and organs. Although numerous transcription factors have been characterized from different organisms, the knowledge of the enhancers responsible for their tissue-specific expression remains fragmentary. Here we use *Ciona* to study the enhancers associated with ten transcription factors expressed in the notochord, an evolutionary hallmark of the chordate phylum. Our results illustrate how two evolutionarily conserved transcription factors, Brachyury and Foxa2, coordinate the deployment of other notochord transcription factors. The results of these detailed *cis*-regulatory analyses delineate a high-resolution view of the essential notochord gene regulatory network of *Ciona*, and provide a reference for studies of transcription factors, enhancers, and their roles in development, disease, and evolution.

The notochord is the quintessential synapomorphy of the chordate phylum. In developing chordates, from tunicate larvae to human embryos, this mesodermal axial structure is necessary for support, patterning and morphogenesis of the body plan[1–3]. In vertebrates, the notochord exchanges patterning cues with the sclerotome, which gives rise to the vertebral bodies; as the vertebral column develops, the notochord regresses and its remnants form the *nuclei pulposi*, the central-most region of the intervertebral discs, which are highly hydrated and possess shock-absorbing properties[4]. Although its evolutionary origins and its relation to analogous structures in non-chordate phyla remain still debated[5,6], the severe effects of notochord ablation[7,8] and malformations[9–12] unequivocally assert the role of the notochord in embryonic development. The widespread pivotal role of the notochord in chordate development renders studies of the gene regulatory network (GRN) underlying its formation both

compelling and broadly informative. Studies of the notochord GRN in vertebrates are often prone to organism-specific limitations, such as slow development, scarce accessibility/visibility of the notochord cells, which become confined to the *nuclei pulposi* during early embryogenesis, genetic redundancy, and the costs and laboriousness of transgenic experiments. Most of these limitations can be overcome through the use of tunicate larvae, invertebrate chordates closely related to vertebrates that develop a tractable notochord within the first day following fertilization[13–16]. In these embryos, notochord development usually involves the formation of two rows of 20 post-mitotic cells that converge to the midline of the embryo and intercalate to form a single rod of 40 cells with a characteristic stack-of-coins arrangement; this process is mediated by the formation of mediolaterally oriented actin-based protrusions[17,18]. After intercalation, notochord cells undergo very extensive changes in

[1]Department of Molecular Pathobiology, New York University College of Dentistry, New York, NY 10010, USA. [2]School of Biological Sciences, Georgia Institute of Technology, Atlanta, GA 30332, USA. [3]Post-Baccalaureate Premedical Program, Washington University, St. Louis, MO 63130, USA. [4]These authors contributed equally: Lenny J. Negrón-Piñeiro, Yushi Wu. ✉e-mail: adg13@nyu.edu

shape and size, as in most species the tail elongates and forms a central fluid-filled lumen[2].

In addition to having a visible and fast-developing notochord, the cosmopolitan tunicate *Ciona robusta* can be easily transfected at the 1-cell stage with multiple plasmids to produce hundreds of synchronously developing embryos, and possesses a compact, fully annotated genome (~140 Mb)[19], with most transcription factors (TFs) present as single-copy genes[20,21] and modular *cis*-regulatory regions usually located in the proximity of the genes that they control[22]. Extensive expression surveys and morpholino oligonucleotide knockdowns of TFs have laid a remarkable foundation for studies of tissue-specific GRNs in *Ciona*[20,23–27] and have highlighted the evolutionary conservation of the main TFs necessary for the formation of its notochord, Brachyury and Foxa2[23,28,29]. These TFs are linked in a cross-regulatory circuit that lies at the core of the notochord GRN[30–34] and controls the expression of other notochord TFs and of scores of effector genes. The *Ciona* notochord GRN coordinates evolutionarily conserved morphogenetic milestones, including cell movements, intercalation, extracellular matrix secretion, cell-shape changes and tail elongation. In *Ciona*, we have uncovered the regulatory connection between Brachyury, which in *Ciona* is notochord-specific[28] (hereinafter abbreviated as Ci-Bra) and two other notochord TFs, Tbx2/3 and Xbp1, which act as Ci-Bra intermediaries[35,36]. We have also presented evidence that the positive feedback between Brachyury and Xbp1, which we first uncovered in *Ciona*, is conserved in *Xenopus* embryos[36]. To further illuminate the regulatory connections that make up the notochord GRN, we selected for the present study ten TFs expressed in the *Ciona* notochord during widely different developmental windows. These TFs are the *Ciona* orthologs of vertebrate TFs that are also expressed in the notochord and/or its derivatives, such as *Lmx1a*, which is expressed in the mouse notochord[37] and in chordoma, a notochord-derived tumor[38]; *Ror-a*, a marker of the mouse *nuclei pulposi*[39]; *Etv1*, which is expressed in the primitive streak and notochord in chick embryos[40–43]; *Spalt-like* (*Sall1*), which has been detected in the notochord of zebrafish, chick and mouse[44–46]; *Aff4*, *CasZ1* and *Islet1*, which are enriched in node/notochord cells of gastrula-stage mouse embryos, according to scRNA-Seq surveys[47]. In addition to these TFs, we analyzed the transcriptional regulation of *Cnot11*, which in other organisms encodes a component of a large complex involved in gene regulation[48], and *Mxd1*, which in different systems is part of a network of TFs that control numerous cellular and developmental processes[49]. Even though most of these TFs were known to be expressed in the notochord and/or *nuclei pulposi* across chordates, along with an increasing number of structural genes[50], no information was available about the *cis*-regulatory regions that control their notochord expression, and their respective positions within the notochord GRN remained to be determined. In *Ciona*, we have shed light on the main features of the notochord CRMs associated with these TFs, and we have identified their respective activators. The information gathered through this study has allowed us to delineate the regulatory circuitry that connects these TFs and coordinates the formation of the structure that characterizes, supports, and patterns the developing body plan of all chordates.

## Results

### Modular *cis*-regulatory regions recapitulate the expression patterns of ten notochord TFs

We identified *cis*-regulatory sequences from the genomic loci of ten TFs expressed in the notochord of *Ciona robusta* (formerly *Ciona intestinalis* type A[51]; hereinafter *Ciona*) during different stages of its development (Table 1). Genomic fragments spanning on average 1-2 kb were selected based on their proximity to the putative transcription start sites of each TF, on the exon/intron structure of each coding region, and on existing data on chromatin accessibility and occupancy by notochord TFs, which were acquired through ATAC-Seq and ChIP-

**Table 1 | *Ciona robusta* notochord TFs analyzed in this study**

| Ciona gene name | Human counterparts and % identity | TF Family | KY2021 gene model | EST | Candidate early activators (Kubo et al.[54]) |
|---|---|---|---|---|---|
| *Aff4* (AF4/FRM2 family member 4) | *AFF4* (32.01%); *AFF3* (30.66%); *AFF2* (21.35%) | AF4/FRM2 | KY21.Chr2.630 | 85P09 | Bra, Foxa.a, ZicL |
| *Cas21* (Castor Zinc Finger 1) | *CAS21* (70.56%) | Zinc-finger C2H2-type | KY21.Chr1.1798 | 75F01 | Bra, ZicL |
| *Cnot11* (CCR4-Not transcription complex subunit 11) | *CNOT11* (86.01%) | Component of the CCR4-NOT complex | KY21.Chr2.616 | 62K09 | |
| *Etv1* (Ets variant transcription factor 1) | *ETV1* (75.40%); *ETV4* (72.44%); *ETV5* (72.44%) | ETS | KY21.Chr1.779 | 30h24 | Foxa.a |
| *Fli1/Erg* (Ets-related gene) | *FLI1* (40.24%); *ERG* (39.74%); *ETS1* (33.33%) | ETS | KY21.Chr4.655 | 29o19 | Foxa.a |
| *Islet1* (ISL LIM homeobox) | *ISL2* (57.18%); *LHX9* (35.88%) | Homeodomain | KY21.Chr4.1164 | 58E06 | Bra, ZicL |
| *Lmx1-r* (LIM homeobox transcription factor 1-related) | *LMX1B* (47.69%); *LMX1A* (41.39%); *LHX5* (35.23%) | Homeodomain | KY21.Chr9.611 | 45i22 | Bra, Sna, ZicL |
| *Mxd1* (MAX dimerization protein 1) | *MXD1* (41.82%); *MXI1* (41.58%); *MXD4* (40.42%) | bHLH | KY21.Chr1.459 | 60E03 | Bra, Foxa.a, Sna, ZicL |
| *Ror-a* (RAR related orphan receptor) | *ROR-A* (45.15%); *ROR-B* (44.79%); *ROR-C* (41.47%) | NR1 subfamily of nuclear hormone receptors | KY21.Chr3.203 | 82E24 | Sna |
| *Spalt-like* | *SALL3* (47.17%); *SALL4* (40.71%) *SALL1* (30.91%) | Zinc-finger C2H2-type | KY21.Chr4.1016 | 23n09 | ZicL |

Seq experiments, respectively[52–55]. Selected genomic regions were cloned upstream of the *Ciona Foxa.a* basal promoter region in a plasmid vector containing the *LacZ* reporter gene[56] and tested for *cis*-regulatory activity in vivo by electroporation in *Ciona* zygotes[15]. Genomic regions that were able to drive reporter gene expression in the notochord were subsequently reduced to 'minimal' notochord *cis*-regulatory modules (CRMs), ranging from ~100 to ~560 bp, and subjected to sequence analysis and site-directed mutagenesis. For the identification of putative binding sites, we used as a reference functional binding sites previously identified within in vivo validated *Ciona* notochord CRMs[35,57–60].

Additionally, the genomic regions corresponding to the *C. robusta* notochord CRMs were cloned from *C. savignyi* and tested in *C. robusta* to assess the conservation of *cis*-regulatory information across these divergent species. In total, 163 constructs were generated and individually tested in vivo. To avoid statistical dispersion, the effects of each truncation/mutation were quantified by scoring the number of transgenic embryos exhibiting notochord staining in ≥3 separate experiments carried out on ≥3 different batches of animals. The genomic coordinates and location of the enhancers identified in this study, and their distances from their respective transcription start sites are reported in Supplementary Table 1.

## Foxa.a presides over a Brachyury-independent branch of the *Ciona* notochord GRN

We had previously reported the identification of *Lmx-like* (gene model: KH.C9.485; recently renamed *LIM homeobox 1-related*, or *Lmx1-r*) and provided evidence of its Ci-Bra-independent notochord expression[25]. This gene is characterized by an early and prolonged expression in the developing notochord, which begins at gastrulation and persists throughout the tailbud stages (Fig. 1A). To identify the *cis*-regulatory region associated with this gene, a 1.2-kb genomic fragment located at the 5′-end of the *Lmx1-r* coding sequence was cloned and tested in vivo (Fig. 1B), and was found to direct *LacZ* expression in notochord and CNS (Fig. 1C). This 1.2-kb region was reduced, through serial truncations, to a 169-bp notochord CRM. Sequence analysis revealed that the 169-bp CRM contains a putative binding site for Brachyury and/or other TFs of the T-box family[35,57,59,61,62], two binding sites for TFs of the Fox family[29,32,58] and a sequence resembling a binding site for TFs of the CREB family[63] (Fig. 1D). The function of these sequences was tested by site-directed mutagenesis (Fig. 1E–J) and by scoring a large number of transgenic embryos, in order to ensure statistical robustness of the sampling (Fig. 1K). This analysis indicated that both the Bra/T-box and CREB binding sites were dispensable, while one of the putative Fox binding sites was necessary for reporter gene expression in notochord cells (Fig. 1D, K). Our previous studies of notochord CRMs indicate that Bra/T-box binding sites with the sequence TAGCAC (Fig. 1D) are seldom found in notochord CRMs, and they are yet to be reported as functional/necessary for notochord activity[57]. Together with the mutagenesis results and the unperturbed expression of *Lmx1-r* in the notochord of *Ci-Bra* mutants[25], these findings support the hypothesis that the expression of *Lmx1-r* does not require Ci-Bra, and confirm the existence of a Brachyury-independent branch of the notochord GRN in *Ciona*.

Next, we retrospectively analyzed the interspecific conservation, chromatin accessibility, and TF occupancy of the 169-bp notochord CRM (Fig. 1L) and its genomic surroundings (Supplementary Fig. 1), taking advantage of the availability of the genome of the congener species *Ciona savignyi* and of publicly available ATAC-Seq, ChIP-on-chip and ChIP-Seq data for *C. robusta*[52,53,55]. VISTA[64] alignments and reciprocal BLASTN searches[65] (Fig. 1L, top panel and Supplementary Table 3), indicate that the 169-bp sequence is highly conserved between *C. robusta* and its distant relative *C. savignyi* (Fig. 1L, top panel; pink peaks represent conserved non-coding sequence), notwithstanding the considerable evolutionary distance between these

species[66]. Nevertheless, while both sequence and location of the dispensable Bra binding site are preserved, the Fox binding site necessary for notochord activity does not appear to be conserved in *C. savignyi*, at least not in the same location (Supplementary Table 3). These observations prompted us to clone a genomic fragment encompassing this region from *C. savignyi*, and to test it in *C. robusta*. We found that the *C. savignyi* genomic fragment is able to direct reporter gene expression in notochord cells, possibly because its sequence contains two putative Fox binding sites (Supplementary Fig. 1).

Chromatin accessibility profiles[52,53] indicate that the 169-bp notochord CRM overlaps with a region of accessible chromatin (i.e., above the peak-calling threshold)[52] (Fig. 1L and Supplementary Fig. 1). In order to identify the TF of the Fox family required for the activity of the 169-bp CRM, we analyzed the available ChIP-on-chip and ChIP-Seq data on TF occupancy[54,55]. In *Ciona*, *Foxa.a* (formerly *forkhead/HNF3beta*; gene model KH.C11.313) is expressed in the developing notochord and its precursors, as well as in CNS and endoderm[29], in a pattern that resembles the expression of its mouse counterpart, *Foxa2*[31]. Chromatin occupancy studies have shown that this TF occupies the genomic loci of >3800 *Ciona* genes in early embryos, including those encoding for various notochord TFs[54]. Another TF of the Fox family transiently expressed in notochord precursors is Foxd, which is an early activator of *Ci-Bra* expression[67] and binds both the *Ci-Bra* and the *Foxa.a* promoter regions in early *Ciona* embryos[54]. For the interpretation of occupancy data, we followed the cut-off established by Kubo et al.[54], and we cross-referenced their ChIP-on-chip data with those available for well-characterized regulatory sequences. In particular, we referred to the *dmrt* promoter region, which is bound in vitro by a GST-Foxd fusion protein[55] and is bound by a Foxd-GFP fusion protein in ChIP-on-chip assays[54], and to the *Ci-tune* notochord CRM, which is bound in vitro by a GST-Bra fusion protein[32] and is occupied by a Bra-GFP fusion protein in ChIP-on-chip assays[54]; this latter region is also bound by a GST-Foxa.a fusion protein[32], and, accordingly, ChIP-on-chip assays display occupancy in 64-cell embryos by a Foxa.a-GFP fusion protein[54]. When these parameters were used, only the occupancy of the 169-bp *Lmx1-r* notochord CRM by Foxa.a appeared significant (Fig. 1L).

To verify the role of Foxa.a in the activation of the *Lmx1-r* notochord CRM, we ectopically expressed this TF in muscle cells and their precursors, using the 737-bp muscle-specific *Snail* promoter[68]. Embryos co-electroporated with the *Sna > Foxa.a::GFP* plasmid[32] along with the 169-bp *Lmx1-r > LacZ* notochord CRM exhibit reproducible ectopic staining in muscle cells (Fig. 1M), providing evidence that Foxa.a is not only required for the activity of this CRM, but is also sufficient to elicit its expression in an ectopic territory. The percent of ectopic staining dropped significantly when the *Sna > Foxa.a* plasmid was co-electroporated with the 169-bp *Lmx1-r > LacZ* CRM carrying a mutation in the Fox binding site (average frequency of cells co-expressing these constructs: <10%; likely due to the leaky activity of both drivers in mesenchymal cells) (Fig. 1N; see Supplementary Fig. 1 for an embryo with a mild phenotype). The ectopic expression of *Foxa.a* in muscle cells and their precursors visibly affected muscle development, and consequently tail extension, with a severity that was directly proportional to the extent of transgene incorporation (Supplementary Fig. 1). In particular, embryos displaying a mild phenotype due to mosaic incorporation of the *Sna > Foxa.a* plasmid developed a partial notochord flanked by small groups of underdeveloped muscle cells, while embryos with higher incorporation of the transgene were characterized by a shorter, stubby tail and a more disorganized notochord (Supplementary Fig. 1). The phenotype we observed is consistent with previous reports of a cross-talk between developing notochord and muscle, which is necessary for proper tail extension[69]. Conversely, in embryos ectopically expressing Bra, *Lmx1-r* expression remained confined to the notochord (see below).

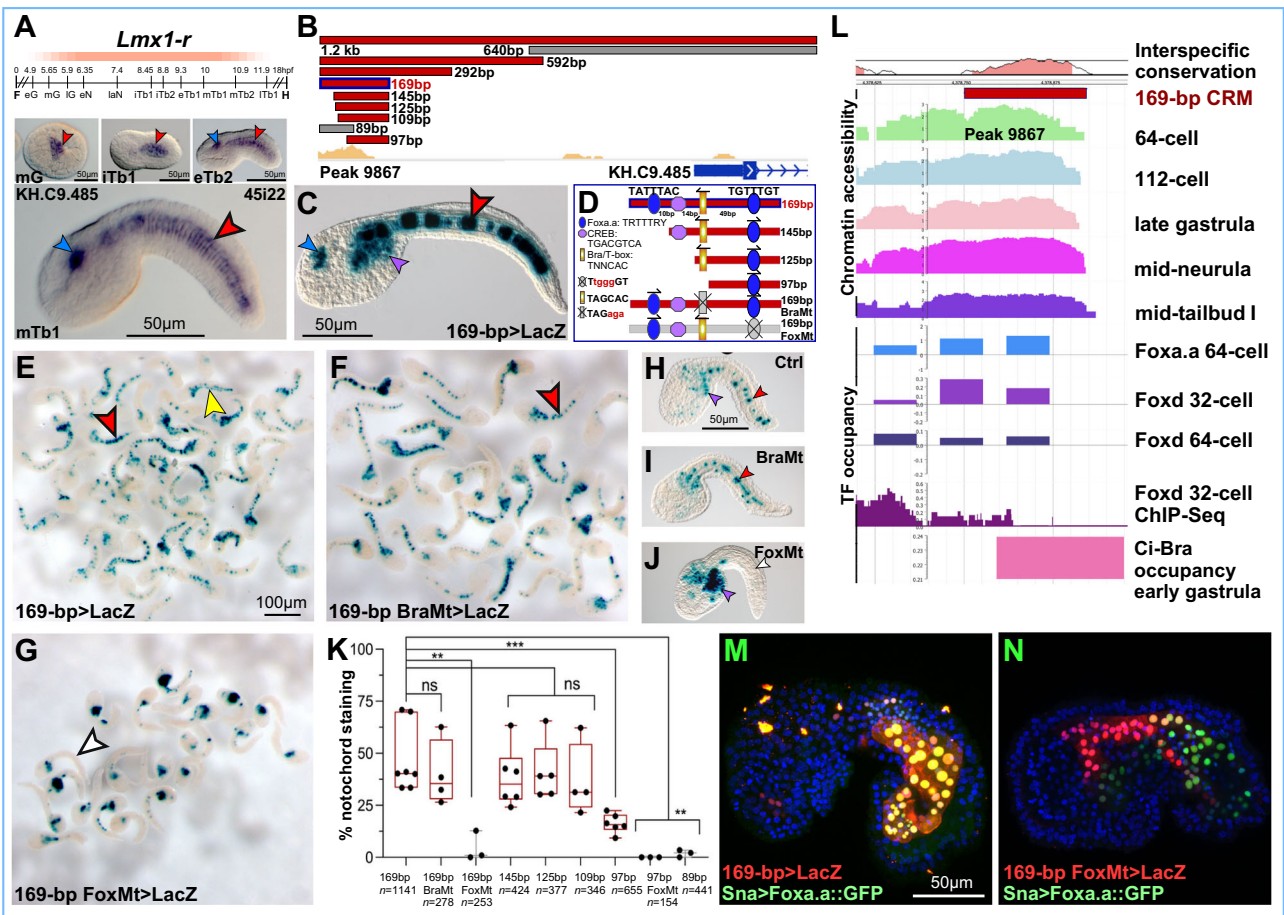

**Fig. 1 | *Lmx1-r* is part of a Brachyury-independent branch of the notochord GRN and is controlled by Foxa.a. A** Expression window (peach-colored bar) of *Ciona robusta Lmx1-r* (KH.C9.485; Table 1), superimposed to the *C. robusta* developmental timeline[134], determined by WMISH using a probe synthesized from EST 45i22. Microphotographs: wild-type *C. robusta* embryos at the developmental stages reported in each panel. eG early gastrula, eN early neurula, eTb early tailbud, F fertilization, H hatching, iTb initial tailbud, laN late neurula, lG late gastrula, lTb late tailbud, mG mid-gastrula, mTb mid-tailbud. **B** Genomic regions tested in vivo (colored horizontal bars; red, notochord activity; gray, no detectable activity), mapped to the *Lmx1-r* gene model (blue horizontal bar; introns represented by lines). ATAC-Seq peaks[52] (yellow ochre) reproduced with permission from Aniseed[131] (https://www.aniseed.cnrs.fr). **C** Mid-tailbud embryo electroporated with the 169-bp region in **B**. **D** Truncation/mutation analysis. Putative TF binding sites are depicted as geometric shapes. Mutations: lower case, red font; mutated binding sites: gray, covered by 'X'. Mt, mutant. **E–G** Low-magnification microphotographs of transgenic mid-tailbud embryos; scale bar in **E**. **H–J** Representative embryos from the experiments in **E–G**; scale bar in **H**. **K** Quantification of the truncation/ mutation analysis. The total number of stained embryos (*n*) analyzed per experiment is reported underneath the *x*-axis. Data represent 25th to 75th percentiles (bounds of box), median (center line) ± min to max (whiskers); **$p < 0.01$, ***$p < 0.001$, ****$p < 0.0001$, ns non-significant, two-sided Student's *t* test ($n \geq 3$ biologically independent samples per category); source data are provided as a Source Data file. **L** Interspecific conservation, accessibility and TF occupancy of the 169-bp notochord CRM (**C**, **D**). Top: Interspecific conservation of the 169-bp CRM and its surroundings, reproduced with permission (VISTA[64] genome browser https://genome.lbl.gov/vista/index.shtml). Middle: Accessibility of the 169-bp CRM, determined by ATAC-Seq (64-cell, 112-cell, late gastrula, mid-neurula[52]; mid-tailbud I[53]). Bottom: Occupancy of the 169-bp CRM by Foxa.a, Foxd and Ci-Bra, determined by ChIP-on-chip and ChIP-Seq[54,55]; reproduced with permission (Ghost genome browser[135] http://ghost.zool.kyoto-u.ac.jp). **M**, **N** Merged confocal microphotographs of embryos carrying the transgenes on the bottom left, immunostained for beta-galactosidase; nuclei stained by DAPI. Scale bar in **M**. Arrowheads color-code: red, notochord; blue, CNS; purple, mesenchyme; yellow, endoderm; white, unstained notochord.

As a next step, we employed CRISPR/Cas9-mediated genome editing of the *Lmx1-r* coding region to begin shedding light on the role of *Lmx1-r* in notochord formation. We used two single-chain guide RNAs (sgRNAs) designed to target *Lmx1-r* exon 2, sgRNA 2.62 and sgRNA 2.108 (Supplementary Fig. 1), along with the developmentally neutral marker *Bra > GFP*[28] (Fig. 2A) and a *Bra > Cas9* plasmid that expresses *Cas9* in the developing *Ciona* notochord (Fig. 2B). Illumina sequencing of target amplicons indicated that the efficiency of the sgRNAs to generate indels is ~19% (sgRNA 2.62) and ~28% (sgRNA 2.108) (Supplementary Fig. 1, see Methods for details). Expression of Cas9 was verified first by WMISH in an independent experiment, and by antibody staining in the knockdown experiments (Fig. 2B–F, insets). Compared to the *Bra > GFP* controls and to embryos carrying only the *Bra > Cas9* transgene, incubated in

parallel, embryos expressing either sgRNA or both displayed a reproducible notochord phenotype (Fig. 2G, H). The notochord cells of *Lmx1-r* CRISPant embryos (Fig. 2C–F) failed to acquire the normal stack-of-coins configuration that is seen in control embryos after intercalation is completed, and this affected both tail shape and elongation. Although most CRISPant notochord cells formed visible medially-oriented protrusions (yellow arrowheads in Fig. 2C', D'), a variable number of cells per embryo exhibited abnormal shapes (orange arrowheads in Fig. 2D'–F').

Together, these results confirm the existence of a circuit of the notochord GRN that is directly controlled by Foxa.a independently of Ci-Bra. The phenotype shown in Fig. 2 suggests that through Lmx1-r, Foxa.a fine-tunes cell-shape changes and intercalation movements required for complete tail elongation.

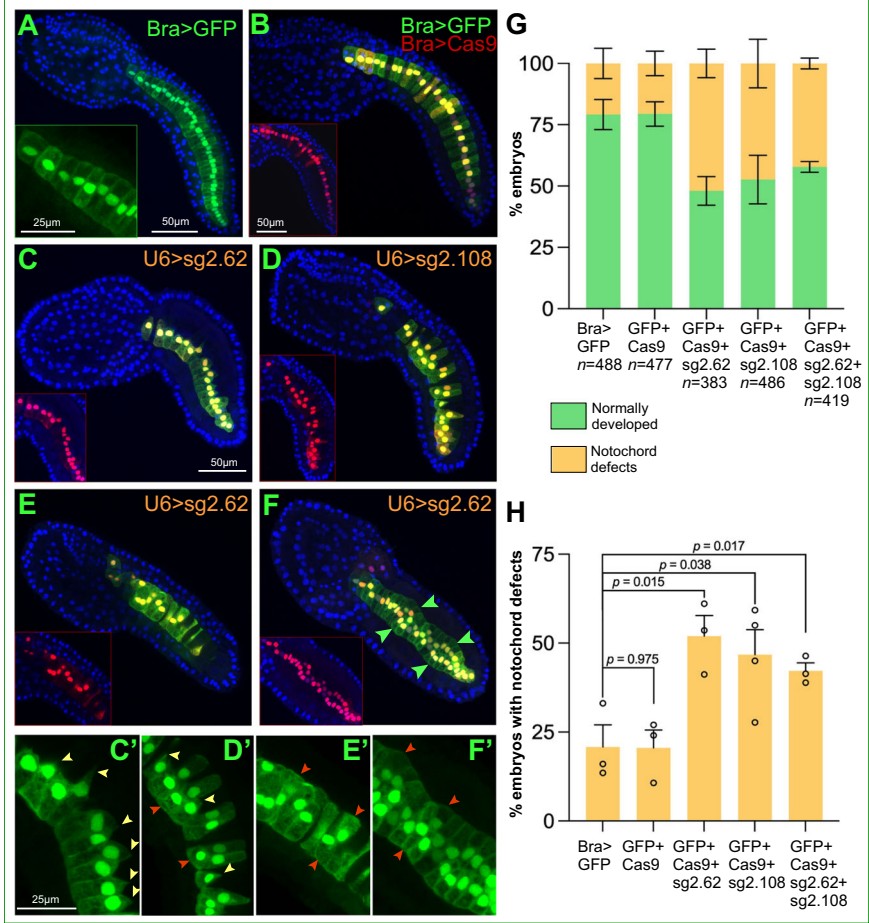

**Fig. 2 | Effects of CRISPR-Cas9-mediated loss of function of *Lmx−1r* on notochord development. A, B** Merged confocal microphotographs of *C. robusta* midtailbud embryos carrying the transgenes indicated on the top right corners. Embryos in **C–F** carry the transgenes shown in **B** in addition to the transgene indicated in the top right corner of each panel. Embryos in **B–F** were immunostained with a Cas9 antibody. Inset in **A** shows notochord cells in a normally developed control embryo for reference. Insets in **B–F** show the merged red and blue channels of the embryos in their respective panels, to monitor expression and nuclear localization of Cas9; scale bar in **B**. Green arrowheads in **F** indicate areas where notochord cells failed to intercalate properly. **C′–F′** High-magnification views of groups of notochord cells from the embryos in **C–F**, respectively; scale bar in **C′**. Yellow arrowheads indicate protrusions; dark orange arrowheads indicate abnormally developed cells. **G** Graph showing the incidence of notochord defects in transgenic embryos cultured in parallel, in triplicate experiments. The error bars represent the standard error of the mean (SEM). The total number of stained embryos (*n*) scored per experiment is shown underneath each construct on the *x*-axis. In each bar, green indicates the fraction of normally developed embryos and yellow labels the fraction of embryos displaying notochord defects. **H** Graph showing a direct comparison of the fractions of embryos displaying notochord defects among the different samples from the graph in **G**. The numbers of embryos scored are reported in **G**. The significance of the two-sided *t*-test is reported in the form of asterisks: (*) *p*-value < 0.05; (**) *p*-value < 0.01; (***) *p*-value < 0.001; ns nonsignificant. Source data are provided as a Source Data file.

## A substantial fraction of the nodes of the *Ciona* notochord GRN is controlled by Brachyury

In a preceding study, we had determined the requirement of Ci-Bra for the notochord expression of *Aff4* (published as *Ci-AFF*[25]) by performing whole-mount in situ hybridization (WMISH) on embryos carrying a null *Ci-Bra* mutation[25,70]. In this study, we have identified and characterized a notochord CRM associated with this gene. *Aff4* (*AF4/FMR2 family member 4*) encodes an elongation factor that acts as a scaffold for different components of the transcriptional super-elongation complex[71,72]. *Ciona Aff4* is expressed in the developing notochord, beginning from the initial tailbud stage until the mid-tailbud stages are reached, at which point its notochord expression declines; conversely, its expression in sensory vesicle and mesenchyme increases[25] (Fig. 3A). The *Aff4* notochord CRM is contained within a 1.117-kb genomic fragment that overlaps with a region of accessible chromatin (ATAC-Seq peaks 1978 and 1610[52]; red bar in Fig. 3B). The analysis of the *Aff4* genomic locus led us to identify also two regions with enhancer activity in tissues other than the notochord (Fig. 3B). Through serial sequence-unbiased truncations, the 1.117-kb notochord enhancer

region was reduced to a 149-bp CRM able to recapitulate the notochord expression of the longer enhancer fragments (Fig. 3B, C). Sequence analysis of this 149-bp CRM revealed the presence of five binding sites for TFs of the T-box family; in *Ciona*, only two members of this family are reportedly expressed in the notochord: Ci-Bra and Tbx2/3[20,28,35,73]. The two 3′-located binding sites were proven dispensable, since when the 149-bp region was truncated to a 102-bp fragment lacking these sites the notochord activity remained unaltered (Fig. 3D). Two of the remaining three Bra/T-box binding sites, organized in an imperfect palindrome, were mutagenized within the 102-bp fragment, but this did not affect the notochord staining (Fig. 3D). Lastly, the individual mutation of the Bra/T-box binding site located near its 5′-end was sufficient to abolish the notochord activity of the 149-bp CRM (Fig. 3D, E; Supplementary Fig. 2; Supplementary Table 2), and yielded the same result within a shorter, 128-bp construct (Supplementary Fig. 2). A putative divergent half-site for TFs of the CREB family was identified by sequence inspection in a longer (168-bp) enhancer region; however, its mutation did not have any affect (Supplementary Fig. 2). Additional intermediate constructs were generated

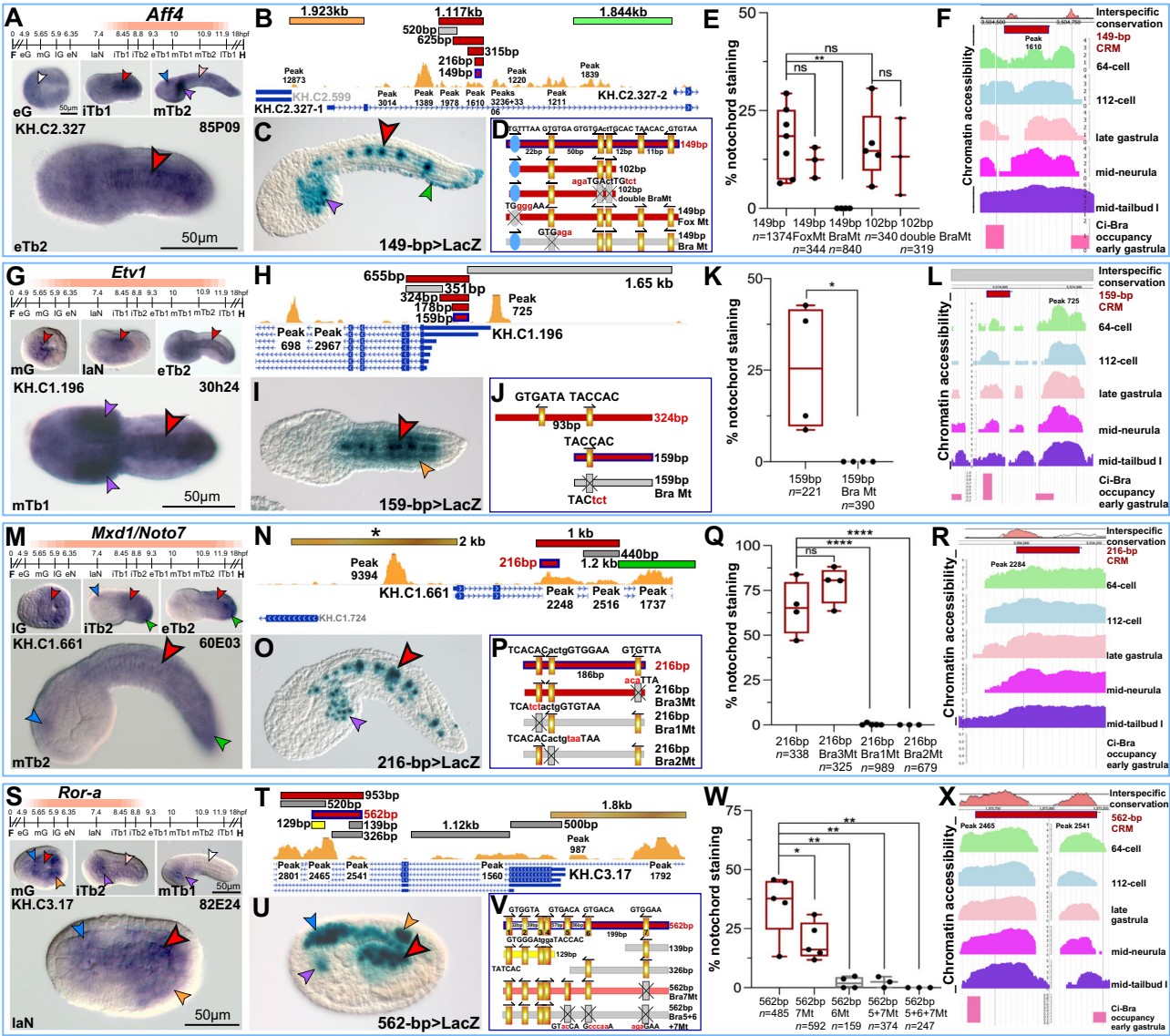

**Fig. 3 | Notochord *cis*-regulatory modules that depend on one or more Bra/T-box binding sites. A, G, M, S** Windows of expression of *C. robusta* notochord TFs (peach-colored bars) superimposed to the *C. robusta* developmental timeline (see Fig. 1), determined through WMISH experiments (Table 1). Microphotographs show wild-type *C. robusta* embryos at the developmental stages reported in the lower left corner of each panel. See Figs. 1 and 2 for scale bars. **B, H, N, T** Identification of notochord CRMs within the genomic loci of the genes in **A, G, M, S**. Horizontal bars represent genomic fragments individually tested in vivo (red, notochord activity, pink, sporadic/weak notochord; gray, no detectable activity; orange, muscle; green, epidermis; brown, multiple tissues); the asterisk in **N** refers to a region previously analyzed[57]. All fragments tested are mapped to regions of accessible chromatin (yellow ochre ATAC-Seq peaks[52]). **C, I, O, U** Embryos carrying the transgenes indicated at the bottom right of each panel. See Fig. 1C for scale bars. **D, J, P, V** The main plasmids used for the identification of the TF binding sites required for notochord activity; horizontal bars color-code: red, notochord activity, pink, sporadic/weak

notochord; gray, no detectable activity. Mutations are in red, lower case. Golden rectangles: putative Bra/T-box binding sites[57] (Supplementary Table 2; Supplementary Fig. 6). Putative binding sites for other TFs that were not analyzed are not depicted. **E, K, Q, W** Quantification of the results of the truncation/mutation analyses. Each bar is the result of 3–5 biological replicates (black dots). *n*, total number of stained embryos analyzed per experiment. In the graphs, two-sided *t*-test significance, whiskers, and other features are as in Fig. 1. Source data are provided as a Source Data file. Mt mutant. **F, L, R, X** Interspecific conservation, accessibility and TF occupancy of the CRMs in **D, J, P**, and **V**. Top: Interspecific conservation of the notochord enhancer regions shown as VISTA[64] plots. In **L**, a gray rectangle indicates lack of VISTA[64] alignment. Middle: Chromatin accessibility of the genomic regions harboring the CRMs shown in **D, J, P**, and **V**, determined by ATAC-Seq (see Fig. 1 for details). Bottom: Occupancy of the genomic regions encompassing the CRMs in **D, J, P**, and **V** by a Ci-Bra-GFP fusion protein[54]. Arrowheads are color-coded as in Fig. 1.

during this analysis, and their respective notochord activity is quantified in Supplementary Fig. 2.

Interspecific sequence comparison showed that even though the 149-bp *Aff4* notochord CRM displays a few short stretches of conserved sequence in *C. savignyi* (Fig. 3F, top) the Bra/T-box binding site necessary for its activity is not retained in the same location in this species (Supplementary Table 3; Supplementary Fig. 2). Consistent with this finding, a genomic fragment encompassing the conserved *C. savignyi* non-coding sequence is unable to direct notochord gene

expression in *C. robusta* (Supplementary Fig. 2). ATAC-Seq profiles suggest that the 149-bp region is accessible during development (Fig. 3F, middle panel; Supplementary Fig. 2). Published ChIP-on-chip data indicate that the region encompassing the *Aff4* notochord CRM is bound by Ci-Bra in embryos at the early gastrula stage (Fig. 3F, bottom).

Notochord expression of *Ciona Etv1* (*Ets variant transcription factor 1*), first reported as *ER81*[20], is detected from mid-gastrula to the tailbud stages (Fig. 3G) and is recapitulated by a few genomic

fragments overlapping the 5′-end of its coding region (Fig. 3H, I). The minimal notochord CRM that we characterized spans 159 bp and contains a single Bra/T-box binding site, which is necessary for reporter gene expression in notochord cells (Fig. 3J, K; Supplementary Fig. 3; Supplementary Table 2). VISTA comparisons did not identify interspecific sequence alignments, likely due to gaps in the genome assemblies used by this software (Fig. 3L top, gray rectangle); in fact, when we aligned the sequences of the *Etv1* loci of *C. robusta* and *C. savignyi* we found that the 159-bp notochord CRM and its corresponding region in *C. savignyi* are 52.2% identical, and the Bra/T-box binding site necessary for notochord activity in *C. robusta* is conserved, although with a different core sequence, in *C. savignyi* (Supplementary Table 3). Accordingly, the *C. savignyi* region corresponding to the *C. robusta* notochord CRM is active in the notochord cells of *C. robusta* (Supplementary Fig. 3). *Etv1* is transcribed through most stages of notochord development (Fig. 3G), and its notochord CRM is adjacent to a region of accessible chromatin (Fig. 3L, middle panel; Supplementary Fig. 3). ChIP-on-chip data indicate that this sequence is bound by Ci-Bra in early gastrula embryos (Fig. 3L, bottom).

Transcripts for *Mxd1* (*Max dimerization protein 1*), first reported as *Noto7*[74], are first detected in notochord precursors in late gastrula embryos, and persist in tailbuds (Fig. 3M). We had reported the presence of a 2-kb muscle/CNS enhancer upstream of this gene[57] (labeled by an asterisk in Fig. 3N); by testing additional genomic regions, identified on the basis of chromatin accessibility, we isolated a 1.2-kb epidermal enhancer and a 1-kb region active in notochord cells (Fig. 3N,O). Through sequence-unbiased truncation analysis, we narrowed the latter region to a 216-bp notochord CRM, which contains three Bra/T-box binding sites, two of which are arranged as a quasi-palindrome (Fig. 3P; Supplementary Table 2). Since mutations in either of these latter binding sites are sufficient to abolish reporter gene expression in the notochord (Fig. 3P,Q; Supplementary Fig. 4), it seems that these half-sites might act as an individual binding site. On the other hand, the mutation of the third Bra/T-box binding site did not affect notochord staining (Fig. 3P,Q). Unexpectedly, the 216-bp Bra2Mt constructs drives *LacZ* expression in the midline epidermis (Supplementary Fig. 4). We retrospectively scanned the mutant sequence (ACACTGTAATAA, shown in reverse orientation for clarity) contained in this construct for TF binding sites and found that it is similar to a Sox8 binding site[75] (AACACTGNAATTGTT). In *Ciona* there are at least two TFs of the Sox family, SoxB1 (gene model KH.C1.99) and SoxC (gene model KH.C7.523) that are expressed in epidermal precursors and in the midline epidermis, respectively; we might have inadvertently created a binding site for one of these TFs in the 216-bp *Mxd1* Bra2Mt construct.

VISTA analysis of the 216-bp CRM and its surroundings (Fig. 3R, top; Supplementary Fig. 4) indicates that the first Bra/T-box binding site (TCACAC, Fig. 3P; Supplementary Table 2) is replaced by one with a different core sequence (TCCCAC) in *C. savignyi*, while the sequence of the second one (TTACAC) and the spacing between these binding sites are completely conserved (Supplementary Table 3). This combination of binding sites was sufficient to elicit notochord staining when a *C. savignyi* genomic fragment encompassing this sequence was tested in *C. robusta* (Supplementary Fig. 4). Analysis of ATAC-Seq peaks indicates overlap between the 216-bp notochord CRM and a region of open chromatin (ATAC-Seq peak 2248[52]; Fig. 3R, middle panel; Supplementary Fig. 4), which is not reportedly occupied by Ci-Bra during early development[54] (Fig. 3R, bottom).

Lastly, *Ror-a* (*Retinoic acid receptor-related orphan receptor alpha*)[20] displays a relatively narrow window of expression, as its transcripts are first detected in notochord and neural precursors in mid-gastrulae, but begin to fade in initial tailbuds and are no longer detectable in mid-tailbuds (Fig. 3S). Our survey of the *Ror-a* locus for regions with *cis*-regulatory activity led to the identification of a 1.8-kb

enhancer region covering the 5′-UTR and upstream sequences, which directs *LacZ* expression in muscle and CNS; additional regions covering the first intron did not show activity above background in vivo (Fig. 3T). The notochord enhancer region was identified by cloning a 953-bp fragment located in the second intron (Fig. 3T). This region was further narrowed to a 562-bp CRM (Fig. 3U), which is enriched in Bra/T-box binding sites (labeled 1–7 in Fig. 3V), four of which, however, are included in a 129-bp region that is active only in endodermal cells (yellow bar in Fig. 3T, V). Hence, we focused the mutation analysis on the Bra/T-box sites 5–7, which are located outside of the 129-bp endodermal enhancer. The inability of the 139-bp and 326-bp fragments to direct *LacZ* expression in the notochord suggested that since these sites were not sufficient to elicit expression when isolated, they could be acting cooperatively[35,57]. The analysis of individual and combined mutations within the 562-bp region confirmed this hypothesis, and showed that Bra/T-box sites 5, 6 and 7 all contribute to notochord gene expression (Fig. 3V, W; Supplementary Table 2).

The interspecific *robusta*/*savignyi* sequence alignment indicates that while this region is conserved, the exact sequence and location of the Bra/T-box binding sites necessary for its enhancer function in *C. robusta* are not maintained in *C. savignyi* (Fig. 3X, top; Supplementary Fig. 5; Supplementary Table 3). Nevertheless, the *C. savignyi* sequence does contain putative Bra/T-box binding sites and has the ability to direct gene expression in the notochord of *C. robusta* (Supplementary Fig. 5). The chromatin accessibility profiles (Fig. 3X, middle panel; Supplementary Fig. 5) and ChIP-on-chip binding data (Fig. 3X, bottom panel) indicate that this 562-bp region is directly controlled by Ci-Bra and/or by its transcriptional intermediary, Tbx2/3, whose consensus binding site is very similar to that of Ci-Bra[76].

We compared all the functional Bra/T-box binding sites identified through this analysis to the well-characterized T-box binding site found in the *Drosophila orthopedia* regulatory region (Supplementary Table 2) and to the consensus binding site for human BRACHYURY (BRA, T, or TBXT) (Supplementary Fig. 6). These comparisons suggest that, in *Ciona*, functional Bra/T-box binding sites are often either non-palindromic (TNNCAC half-sites) or weakly palindromic, a conclusion that is supported by in vitro mobility assays[61]. Bra proteins from zebrafish, frog, and mouse have been reported to bind half-sites as well[77–79] and it is likely that this might be the case for human BRA, although functional binding sites for this TF in the context of human notochord enhancers are yet to be elucidated.

## Trans-activation assays corroborate the results of *cis*-regulatory analyses

As in the case of *Lmx1-r*, we employed an in vivo *trans*-activation assay to verify the results obtained through the *cis*-regulatory analysis, taking advantage of the notochord-specific expression of *Ci-Bra*[28] and of a Ci-Bra-specific antibody developed in our lab[57,80]. We used the 2.6-kb *Foxa.a* promoter region to drive ectopic expression of *Ci-Bra* in endoderm and CNS precursors (*Foxa.a>Bra* plasmid[81]) and verified the expression of the Ci-Bra protein and the incorporation of the CRMs fused to the *LacZ* reporter through double immunostaining with the Ci-Bra antibody (Fig. 4A, D, G, J, M) and a beta-galactosidase antibody (Fig. 4B, E, H, K, N), respectively. The colocalization of the Ci-Bra protein and each enhancer, as well as the effect of Ci-Bra ectopic expression on the empty vector used in these experiments, were assessed by merging the green and red channels of each confocal image (Fig. 4C, F, I, L, O). The Ci-Bra protein was specifically localized to all the nuclei of the 40 notochord cells in control embryos (Fig. 4A–C, G–I), and to the nuclei of a much larger number of cells in embryos carrying the *Foxa.a > Bra* transgene (Fig. 4D–F, J–O), an indication that the ectopic expression of Ci-Bra in neural and endodermal precursors was successful. Embryos ectopically expressing Ci-Bra in neural and endodermal precursors display a peculiar and highly reproducible phenotype, characterized by the presence of a large mass

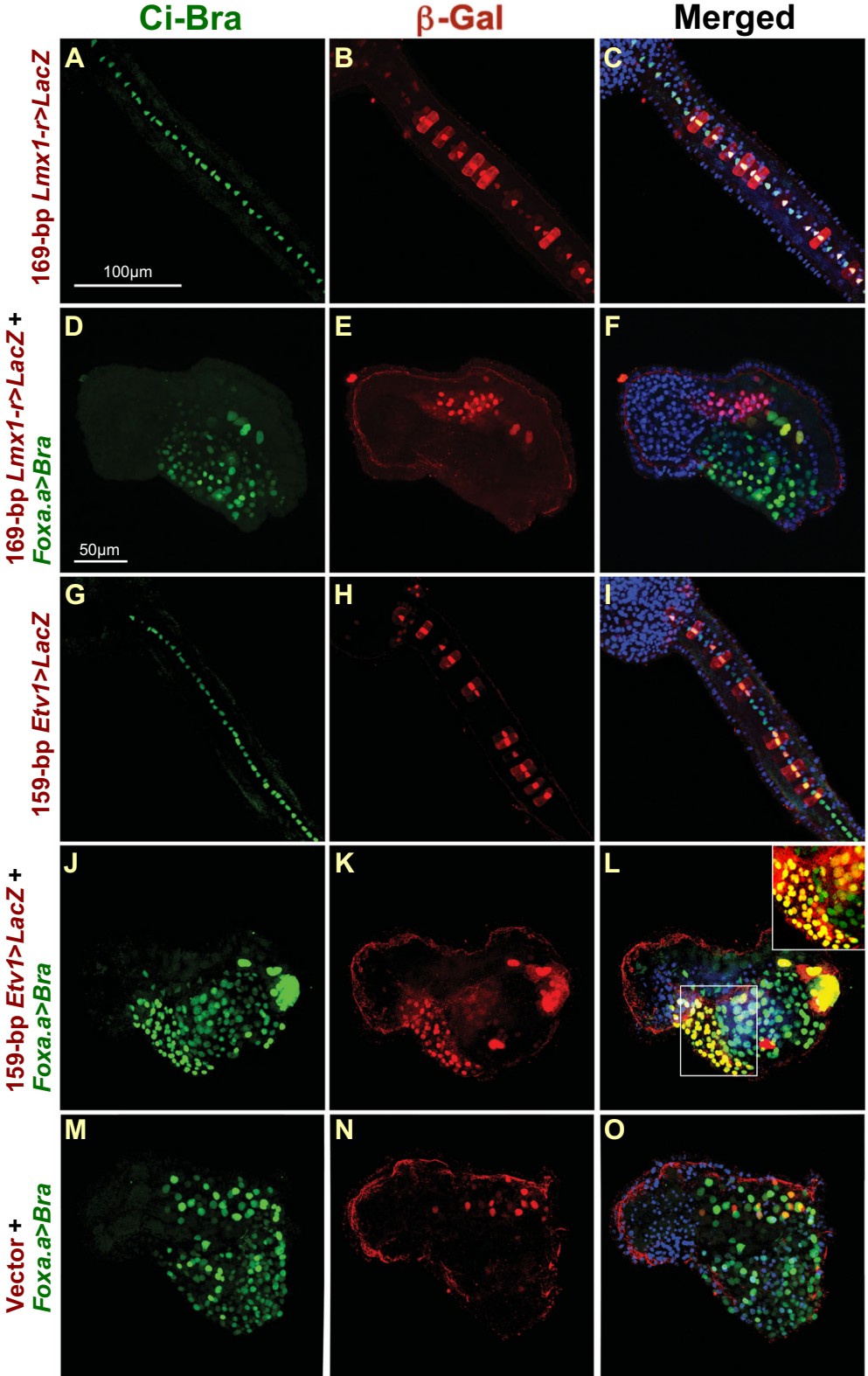

of ventrally located cells[74]. In embryos carrying the *Foxa.a > Bra* transgene and the 169-bp *Lmx1-r* notochord CRM, expression of the *Lmx1-r* CRM remained entopic, limited to the notochord cells and to a small cluster of mesenchymal cells in the trunk (Fig. 4D−F; Supplementary Fig. 8). Conversely, in embryos carrying the *Foxa.a > Bra* transgene and the 159-bp *Etv1* notochord CRM, an overlap in the territories of expression of Ci-Bra, whose ectopic expression was driven by the pervasive 2.6-kb *Foxa.a* promoter[29], and beta-galactosidase, whose expression was driven by the 159-bp CRM, was observed, whereby transgenic embryos displayed the characteristic phenotype induced by the ectopic expression of Ci-Bra and showed ectopic beta-galactosidase staining (86.7% of embryos in a representative experiment; Fig. 4J−L; inset in **L**). The ectopic staining of the 159-bp *Etv1* notochord CRM was more apparent in beta-galactosidase X-gal assays

**Fig. 4 | *Trans*-activation assays in embryos ectopically expressing Ci-Bra.**
Double immunofluorescence experiments performed on *C. robusta* late-tailbud embryos carrying the transgenes indicated on the left side of each row. Confocal images of embryos immunostained with the Ci-Bra antibody[57,80] and a beta-galactosidase antibody. **A**, **D**, **G**, **J**, **M** Green channel images, showing the localization of the Ci-Bra protein in control embryos (**A**, **G**) and in embryos ectopically expressing Ci-Bra in CNS and endoderm (**D**, **J**, **M**). **B**, **E**, **H**, **K**, **N** Red channel images, showing the incorporation of the plasmids containing notochord CRMs upstream of the *LacZ* reporter. Note that not all notochord cells are fluorescent, due to mosaic incorporation of the transgenes. **C**, **F**, **I**, **L**, **O** Images obtained by merging the red, blue (DAPI) and green channel microphotographs of the embryos in **A**, **D**, **G**, **J**, and **M**. Inset in **L** shows a merge of the red and green channels of the region boxed in the main panel, to better highlight the overlapping expression of Ci-Bra and beta-galactosidase in a group of cells (see text). **M**–**O** Control experiment showing the double immunostaining of the pFBΔSP6 vector[56]. This vector per se elicits only sporadic staining in a few muscle and mesenchymal cells (**N**). Images in **A**–**C**, **G**–**I** show higher magnification views of the tail, centered on the notochord; images in **D**–**F**, **J**–**O** show whole embryos, to display the severely altered body plan. Scale bar in **A** is shared with **B**, **C**, **G**–**I**; scale bar in **D** is shared with **E**, **F**, **J**–**O**.

(Supplementary Fig. 7). This response is specific to the *Etv1* notochord CRM, as the vector backbone is unresponsive to the ectopic expression of Ci-Bra (Fig. 4M–O; Supplementary Fig. 7). Similar experiments were carried out for the remaining notochord CRMs directly controlled by Ci-Bra, and yielded comparable results (Supplementary Fig. 7). The results of the co-electroporation of the *Ror-a* enhancer region with the *Foxa.a > Bra* construct were not clear, due to the widespread expression of this CRM in various tissues, which rendered indiscernible any signal that could have been due to ectopic expression.

Together with our previous studies[25], these results confirm that the notochord expression of *Lmx1-r* is independent of Ci-Bra, and that Ci-Bra is not able to elicit ectopic expression of *Lmx1-r*. Instead, Ci-Bra is required for the notochord expression of *Aff4*[25] and for the activity of its notochord CRM, and is sufficient to evoke the ectopic expression of the *Aff4*, *Mxd1*, and other CRMs (Supplementary Fig. 7; see below).

## Transcriptional activators of different families control a subset of the notochord CRMs associated with *Ciona* TFs

Five notochord TFs with widely different temporal onsets, intensity and duration of expression are shown in Fig. 5. The TF with the broadest window of notochord expression is *CasZ1* (*Castor Zn-finger 1*)[20], whose transcripts are first detected in the notochord precursors of initial gastrulae (Fig. 5A). Expression persists in the notochord, CNS, muscle and palps, until the late tailbud stages[20] (Fig. 5A). To identify the *CasZ1* notochord CRM, we cloned different regions encompassing the predicted first introns of different gene models, and during this process we identified separate notochord and muscle enhancer regions (Fig. 5B,C). Through truncation analysis, the notochord CRM was narrowed down to a 128-bp region, containing putative binding sites for TFs of the Zn-finger, Fox, AP-1 and Myb families but devoid of Bra/T-box binding sites (Fig. 5D). Mutations of the putative Zn-finger binding site did not reduce notochord staining (Fig. 5E and Supplementary Fig. 8); however, mutation of the Fox binding site significantly reduced the number of embryos showing notochord staining, while the mutation of the AP-1 binding site abolished notochord staining completely (Fig. 5D, E). These results indicate that this notochord CRM is activated by both Fox and AP-1 family TFs.

VISTA analysis revealed that this regulatory region is highly conserved (Fig. 5F, top; Supplementary Fig. 8); however, its Fox site seems to have been lost in *C. savignyi*, while the sequence of the indispensable AP-1 binding site is maintained (Supplementary Table 3). This change might explain the inability of the corresponding *C. savignyi* sequence to drive notochord activity in *C. robusta* (Supplementary Fig. 8). ATAC-Seq profiles indicate that the 128-bp fragment with notochord activity overlaps with a region of open chromatin (Fig. 5F, middle panel; Supplementary Fig. 8). Even though ChIP-on-chip results indicate that this CRM might be bound by Foxd in early embryos (Fig. 5F, bottom panel), the occupancy score for Foxd in this region is below the threshold that was set for inclusion in the Foxd targets[54,55].

*Cnot11* (*CCR4-NOT transcription complex, subunit 11*), which encodes a subunit of the carbon catabolite repression 4 negative on TATA-less (CCR4-Not) complex, a multifunctional complex involved in various aspects of gene regulation[48], was selected for its narrow window of expression in the notochord, which is detected from early neurula to the mid-tailbud stages (Fig. 5). A 5′-located 1.5-kb region encompassing an area of accessible chromatin displayed notochord activity in vivo (Fig. 5H). This region was reduced to a 158-bp CRM (Fig. 5I) that contains three Bra/T-box binding sites, two AP1 core binding sites, and one Fox binding site (Fig. 5J). Individual mutations of the three Bra/T-box binding sites indicated that only one of them is required for notochord activity. Furthermore, the combined mutation and truncation analyses demonstrated that the Fox binding site is also necessary (Fig. 5J, K; Supplementary Fig. 9). The interspecific conservation of this CRM is limited to short stretches of non-contiguous sequence, and neither one of the binding sites necessary for notochord activity is conserved in *C. savignyi* (Fig. 5L, top; Supplementary Fig. 9 and Supplementary Table 3). This lack of conservation likely explains the inactivity of the *C. savignyi* sequence in the notochord of *C. robusta* (Supplementary Fig. 9). The *Cnot11* notochord CRM maps to a region of accessible chromatin (Fig. 5L, middle and bottom; Supplementary Fig. 9), which according to ChIP-on-chip results is bound by Foxa.a in early embryos below the cut-off established in Kubo et al.[54] (Fig. 5L, middle and bottom); this suggests that the Fox binding site present in this CRM might be bound by a Fox protein other than Foxa.a.

As previously reported[82], *Islet1*, which encodes a TF of the homeobox family, is first detected in palps and CNS, while expression in notochord cells begins around the early tailbud stage and persists throughout the late tailbud stages (Fig. 5M). The *cis*-regulatory sequences that, combined, are able to recapitulate the expression of this gene are contained in two published genomic regions[83,84], indicated by asterisks in Fig. 5N. We first determined that a 140-bp sequence straddling these large genomic regions, and contained in both, was sufficient to direct robust reporter gene expression in the notochord (Fig. 5N); then, we further narrowed this sequence to a 125-bp CRM (Fig. 5O). Through sequence analysis, we identified two adjacent HD binding sites, two Bra/T-box sites and an E-box (CANNTG), the binding site for TFs of the bHLH family[85,86] (Fig. 5P). Individual site-directed mutations of these binding sites revealed that the notochord activity of this CRM relies upon both the HD binding site(s) and the Bra/T-box site adjacent to it (Fig. 5P, Q; Supplementary Fig. 10). Even though VISTA alignments were not available for this sequence (Fig. 5R top, gray rectangle), through reciprocal BLASTN searches and manual alignments we identified a region showing 59.2% interspecific sequence identity and conservation of both the homeodomain (HD) binding site(s) and the Bra/T-box site between *C. robusta* and *C. savignyi* (Supplementary Table 3). When tested in *C. robusta*, this *C. savignyi* region was able to direct reporter gene expression in the notochord (Supplementary Fig. 10). The 125-bp *Islet1* notochord CRM maps to a chromatin region that appears accessible in mid-tailbud stages in whole embryos (Fig. 5R, middle panel)[53] (Supplementary Fig. 10), which is consistent with the late onset of notochord expression of this gene (Fig. 5M). Although ChIP-on-chip assays indicate that this region might be contacted by Fox proteins during early embryogenesis (Fig. 5R), no evident canonical Fox binding sites could be found in this notochord CRM, and this gene is not included in the list of Foxa.a ChIP targets[54]; instead, the occupancy of this locus by Ci-Bra in early embryos is above the threshold established in Kubo et al.[54], lending support to our results.

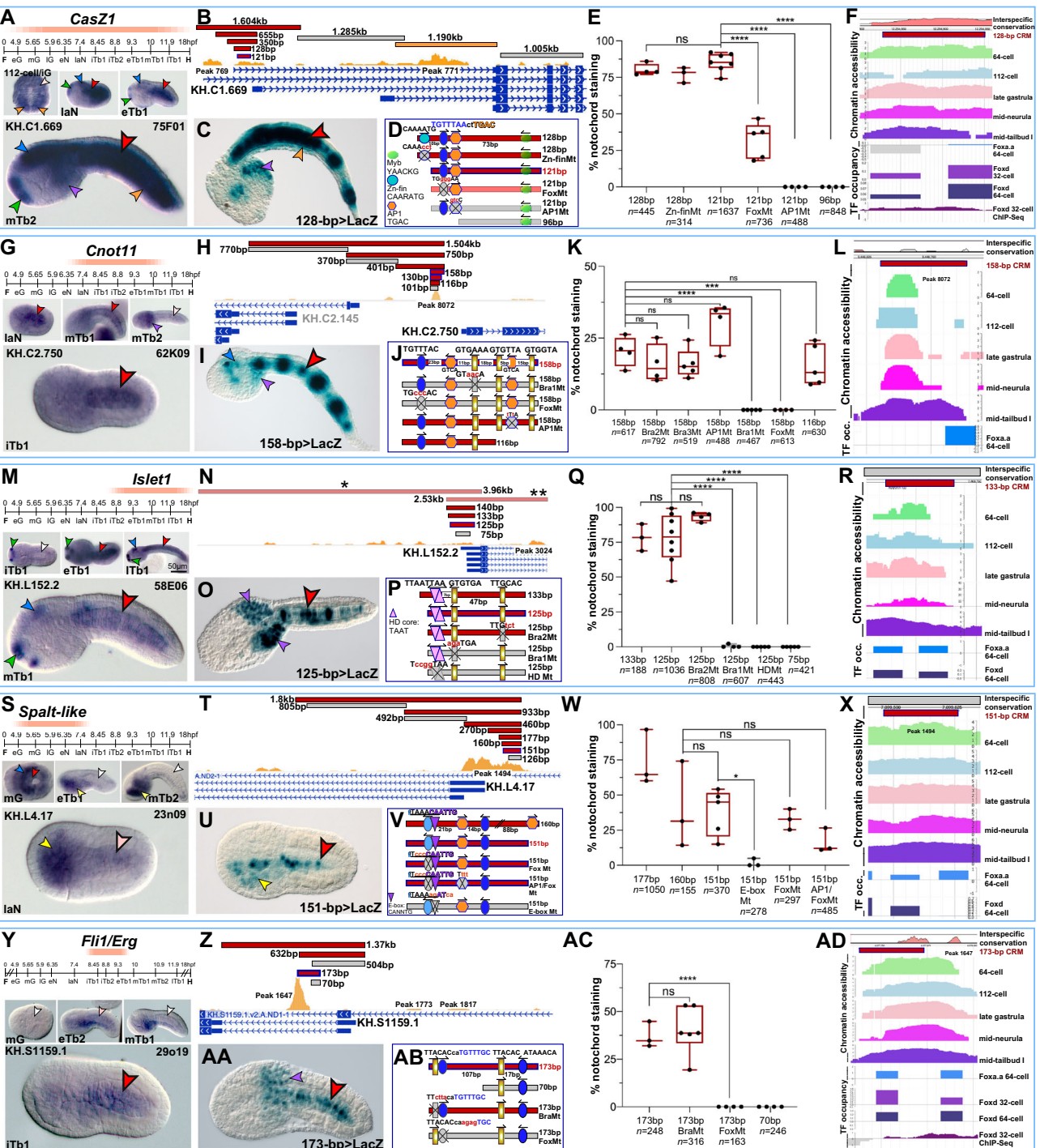

**Fig. 5 | Notochord *cis*-regulatory modules that depend on binding sites for transcriptional activators of different families. A, G, M, S, Y** Windows of expression (peach-colored bars) of *C. robusta* notochord genes encoding TFs (abbreviations are as in Fig. 1), determined through WMISH. ESTs used for RNA synthesis are reported on the upper right corners (Table 1). Microphotographs: wild-type *C. robusta* embryos hybridized in situ at the developmental stages reported in the lower left corner of each panel. **B, H, N, T, Z** Identification of notochord CRMs within the genomic loci of the genes in **A, G, M, S,** and **Y.** Genomic regions of variable length were individually tested in vivo (horizontal bars, color-coded as in Fig. 1); the asterisks in **N** refer to regions that had been previously analyzed (one asterisk[83]; two asterisks[84]). All fragments tested are mapped to predicted gene models and regions of accessible chromatin (yellow ochre ATAC-Seq peaks[52]). **C, I, O, U, AA** Transgenic embryos carrying the plasmids indicated at the bottom right corner of each panel. **D, J, P, V, AB** Truncation analysis and site-directed mutagenesis. Genomic regions are symbolized by horizontal bars, color-

coded as in Fig. 1. Mutations are in red, lower case. **E, K, Q, W, AC** Results of the truncation/mutation analyses. Each bar is the result of 3–7 biological replicates (black dots). The total number of stained embryos (*n*) analyzed per experiment is shown underneath the *x*-axis; two-sided *t*-test significance, whiskers, and other features are as in Fig. 1. Source data are provided as a Source Data file. Mt mutant. **F, L, R, X, AD** Interspecific conservation, accessibility and TF occupancy of the notochord CRMs in **D, J, P, V,** and **AB.** Top: VISTA plots of the regions with notochord activity that were used for truncation/mutation analyses (red horizontal bars)[64]. In **R** and **X** gray rectangles indicate lack of VISTA alignment. Middle: Chromatin accessibility landscapes of the CRMs in **D, J, P, V,** and **AB,** determined by ATAC-Seq (64-cell, 112-cell, late gastrula, mid-neurula[52]; mid-tailbud I[53]). Bottom: Occupancy of the regions of the genomic loci harboring the CRMs in **D, J, P, V,** and **AB** by Foxa.a and Foxd fusion proteins, determined by ChIP-on-chip[54] and by Foxd, determined by ChIP-Seq[55]. Arrowheads are color-coded as in Fig. 1. Scale bars are as in Figs. 1 and 3.

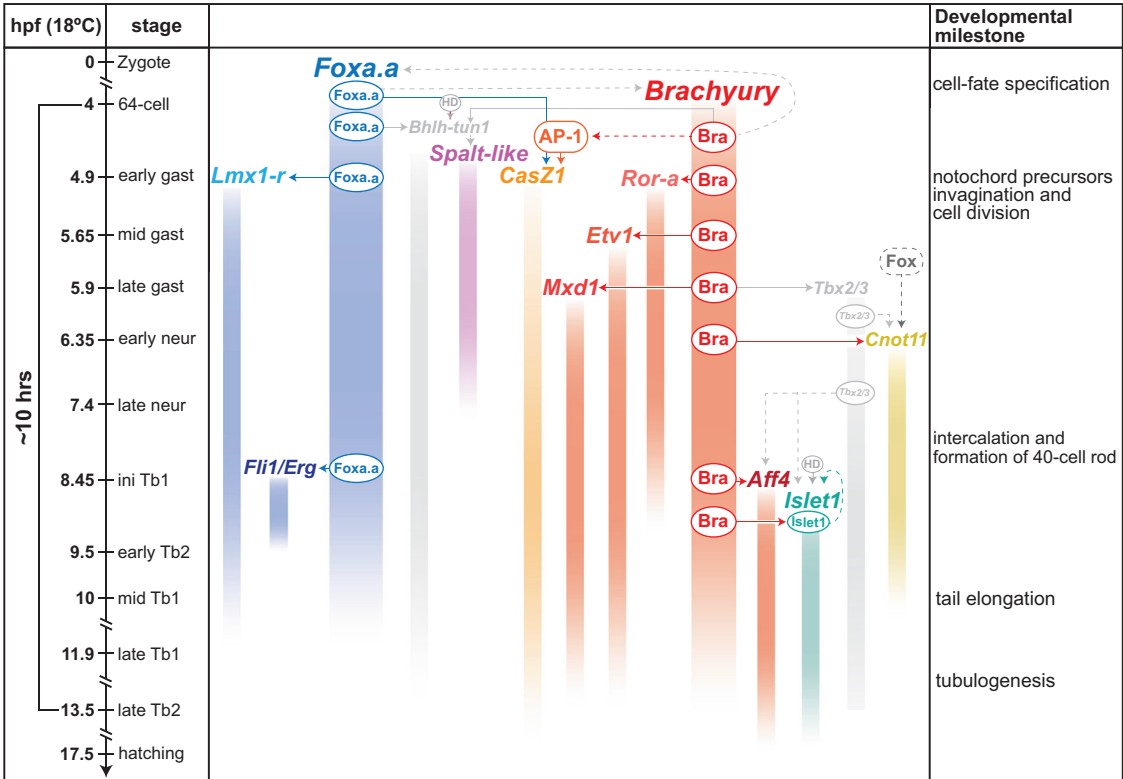

**Fig. 6 | View of the *Ciona* notochord GRN provided by the integration of *cis*-regulatory analyses and *trans*-activation assays.** Left side: Time-table of *Ciona* developmental stages from zygote to hatching larva[134]. Middle: detailed view of the *Ciona* notochord GRN gained through this study. TF proteins are symbolized by ovals, lines indicate regulatory interactions, either direct (solid color) or inferred (dashed). Previously published regulatory interactions and TFs are shaded in gray. Genes on the left side, represented as blue vertical bars, are part of the Foxa.a-downstream branch of the GRN. The genes symbolized by bars colored in different shades of red/orange, *Ror-a*, *Etv1*, *Mxd1* and *Aff4*, are controlled by Ci-Bra (here indicated as Brachyury/Bra). Importantly, *Aff4* and *Cnot11* are expressed after Tbx2/3[35] (gray) and therefore may be activated by Ci-Bra through this T-box TF, which is directly controlled by Ci-Bra through a notochord CRM similar to those

identified here[35]. *Spalt-like*, which requires an E-box for the notochord activity of its CRM could be activated by Bhlh-tun1[85] (gray), and thus, indirectly, by Ci-Bra and Foxa.a; *Cnot11* receives regulatory inputs from both a Fox family TF and from Brachyury and/or Tbx2/3. *Islet1* (green) requires a homeodomain (HD) TF along with Ci-Bra and/or Tbx2/3; this binding site could mediate autoregulation. The *CasZ1* notochord CRM requires binding sites for TFs of the Fox and AP−1 family. The response of the *CasZ1* notochord CRMs to the ectopic expression of Ci-Bra suggests that the expression of its AP-1 activator might be influenced by Ci-Bra (dashed red line). Candidate HD activators for *Bhlh-tun1* and *Islet1* are discussed in the text. Right side: list of the main morphogenetic milestones that punctuate notochord development and differentiation in *Ciona*. Detailed descriptions of these processes can be found in Jiang and Smith[2].

*Spalt-like*, which encodes for a Zn-finger TF, is expressed very early during notochord development, as a faint hybridization signal is detected in notochord precursors beginning at the 64-cell stage[25]; however, around the mid-neurula stage, notochord expression fades and is no longer detectable in tailbuds, while the endodermal expression increases in intensity and persists in both trunk endoderm and endodermal strand (Fig. 5S). We first identified a 1.8-kb fragment with notochord enhancer activity (Fig. 5T) and through serial truncations we narrowed it to a 151-bp notochord CRM (Fig. 5T, U); sequence analysis demonstrated that this region contains a non-canonical Fox binding site (TGTTTAA; light blue in Fig. 5V), one E-box, an AP-1 binding site, and a canonical Fox site. We selected point mutations that specifically affected either the non-canonical Fox binding site or the E-box, and found that the binding site required for notochord activity was the E-box (Fig. 5V, W; Supplementary Fig. 11). VISTA alignments did not highlight interspecific conservation of this CRM (Fig. 5X top, gray rectangle), however, through sequence alignments, we determined that while most of the 151-bp notochord CRM sequence is conserved (78.1% sequence identity), the E-box site is not found in the corresponding location in *C. savignyi* (Supplementary Table 3). Nevertheless, the *C. savignyi* genomic fragment encompassing this region is able to direct notochord gene expression in *C. robusta*, possibly because this sequence contains another E-box with a different core (CACATG). The *Spalt-like* notochord CRM maps to a region of open

chromatin (Fig. 5X, middle; Supplementary Fig. 11); binding was detected in early embryos for both Foxa.a and Foxd (Fig. 5X, bottom), as well as for Ci-Bra[54].

Among the TF genes analyzed in this study, *Fli1/Erg* (*Friend leukemia integration1/Ets-related gene*) displays the narrowest window of notochord expression, spanning approximately the initial- to early-tailbud stage interval (Fig. 5Y). We identified a notochord CRM associated with this gene by testing a 1.37-kb genomic region straddling the predicted first exon and first intron of two of the three gene models available (Fig. 5Z), and we reduced this region to a 173-bp notochord CRM (Fig. 5AA). This sequence contains two Bra/T-box and two Fox binding sites (Fig. 5AB); since one Bra/T-box and one Fox binding site were included in a 70-bp region with no regulatory activity (gray bar in Fig. 5AB), we focused on mutating the remaining Bra/T-box and Fox binding sites outside of this negative fragment, which are separated by only 2 bp. Through mutations that specifically affected each binding site, we found that the Bra/T-box was dispensable, while the Fox binding site was necessary for enhancer activity in notochord cells (Fig. 5AB, AC). VISTA analysis of the 173-bp CRM revealed that the Fox binding site is conserved, although with one permutation, in *C. savignyi* (Fig. 5AD, top; Supplementary Table 3). Accordingly, the *C. savignyi* genomic region containing this sequence functions as a notochord enhancer in *C. robusta* (Supplementary Fig. 12). As mentioned above, the *Fli1/Erg* notochord CRM overlaps a region of open

chromatin (Fig. 5AD, middle; Supplementary Fig. 12). Occupancy data indicate that this CRM is bound by Foxa.a, while the binding by Foxd is below the cut-off set by Kubo et al.[54] (Fig. 5AD, bottom).

To test for any possible regulatory connections of these TFs with Ci-Bra, we co-electroporated their notochord CRMs with the *Foxa.a > Bra* plasmid and evaluated their response to the ectopic expression of Ci-Bra (Supplementary Fig. 7). As expected from the results of the *cis*-regulatory analysis, we observed pervasive ectopic expression of the *Islet1* notochord CRM (Supplementary Fig. 7), which suggests that the TF of the HD family that is required for its activity is either downstream of Ci-Bra or is also broadly expressed in other territories in addition to the notochord. Surprisingly, we detected a similar response in the case of the *CasZ1*, *Cnot11* and *Spalt-like* notochord CRMs (Supplementary Fig. 7), which suggests that even these CRMs might be indirectly controlled by Ci-Bra through transcriptional intermediaries of the families that we have identified through the *cis*-regulatory analysis. However, in the case of *CasZ1* and other CRMs that produce staining in other territories in addition to the notochord, further analysis will be required to ascertain the amount of the response to ectopically expressed Ci-Bra[87]. The results of our transactivation assay indicate that, similarly to *Lmx1-r*, the *Fli1/Erg* notochord CRM is unresponsive to the ectopic expression of this TF (Supplementary Fig. 7).

Lastly, we searched for sequence motifs that might be shared by the CRMs identified in this study. We compared all the "full-length" and all the "minimal" regions active in the notochord using primarily the MEME software (https://meme-suite.org/meme/tools/meme)[88]. This search identified one statistically significant motif, which is enriched in the 1.8-kb *Spalt-like* enhancer region (10 occurrences) and is present in different iterations in the *Islet1*, *Cnot11* and *CasZ1* enhancer regions (Supplementary Fig. 13). This motif, TRTGACGTCA, is related to the binding sites for TFs of the AP-1[89], Atf1/CREB[90] and Xbp1[75] families. Interestingly, a shorter version of this motif, TGACGTCA, is present in the 169-bp *Lmx1-r* notochord CRM (Fig. 1D). In the *Cnot11* notochord CRM, this motif encompasses the core TGAC AP-1 binding site necessary for notochord activity (Fig. 5J, K, Supplementary Fig. 13); however, in the remaining enhancer regions the motif maps outside of the minimal CRM sequences (Supplementary Fig. 13), suggesting that the functional relevance of this sequence might be variable and possibly context-dependent.

## Discussion

### Deconstructing *cis*-regulatory information to reconstruct the *Ciona* notochord GRN

We have taken advantage of the experimental strengths and ideal phylogenetic position of *Ciona*, a 'simple' chordate evolutionarily related to vertebrates, to carry out an unprecedented systematic analysis of ten of the *cis*-regulatory interfaces that compose its streamlined notochord GRN. Within the compact *Ciona* genome, *cis*-regulatory elements are usually adjacent to the coding regions that they control, or embedded within their introns; the activity of dozens of *Ciona* enhancers has been analyzed using truncation/mutation methods and has been ascribed to either a single TF binding site or to a small number of binding sites for one or more TFs[57,58,60,91]. High-throughput studies carried out using barcoded synthetic enhancer variants have tested the effects of several hundreds of thousands mutations on a few known enhancers[92], and seem to confirm this advantageous feature of the *cis*-regulatory regions of *Ciona*, which, along with the low redundancy of genes encoding TFs, renders the identification of transcriptional activators rapid and straightforward.

By deciphering the regulatory information encoded in notochord CRMs associated with evolutionarily conserved TFs, the research presented here has reconstructed the regulatory relationships that connect different nodes of the fast-deploying notochord GRN of *Ciona*. The results of this study are summarized in Fig. 6. Taken together, the findings from the *cis*-regulatory analyses and the results of the *trans*-activation assays indicate that, together with *Lmx1-r*, *Fli1/Erg* is part of a Ci-Bra-independent branch of the notochord GRN. On the other hand, *Etv1*, *Mxd1* and *Ror-a* are directly controlled by Ci-Bra, while *Aff4* is either controlled by Ci-Bra directly or, given its late onset, could be controlled by Ci-Bra indirectly through Tbx2/3. *CasZ1* and *Spalt-like*, whose notochord CRMs are responsive to the ectopic expression of Ci-Bra even though they are devoid of functional Bra/T-box binding sites, are seemingly controlled by this TF indirectly (Fig. 6).

One of the main findings of this study is the direct evidence of a Brachyury-independent branch of the notochord GRN that is likely conserved during chordate evolution. The analysis of the notochord CRM associated with *Lmx1-r*, published occupancy data, and the results of our *trans*-activation assays indicate that the notochord expression of this gene is controlled by Foxa.a. In *Ciona*, *Foxa.a* is expressed in notochord, CNS and endoderm, similarly to its vertebrate counterparts[29,31], while *Lmx1-r* is only expressed in the notochord and in a small region of the sensory vesicle[25]. One explanation for this difference in the expression patterns of these TFs could be that the activation of *Lmx1-r* expression by Foxa.a is counteracted in the endoderm and most of the CNS by tissue-specific transcriptional repressors. This hypothesis is supported by the ectopic endodermal staining that is detected in transgenic embryos carrying truncated versions of the *Lmx1-r* notochord CRM, a pattern that can be attributed to the removal of binding sites for transcriptional repressors. Indeed, previous findings have determined that transcriptional repression is required for cell-fate specification in *Ciona*[93], and to maintain Ci-Bra expression confined to the notochord[94].

Through CRISPR/Cas9-mediated genome editing of *Lmx1-r* in the notochord, we have gained a first insight into the function of this TF, which seems likely to be mainly involved in the regulation of the shape and movements of developing notochord cells. As it could be expected, the loss of function of *Lmx-1r* does not cause the dramatic defects caused by the loss of function of *Ciona Brachyury*[70]; *Lmx1-r* CRISPant notochord cells are often misshapen, nevertheless they are able to arrange themselves into rows and to extend mediolateral protrusions. It remains to be determined whether these protrusions are fully functional, since, for example, in *Ciona fibronectin* (*Ci-Fn*) CRISPant embryos protrusions are formed but are unable to lead the notochord cells to intercalate properly, and are eventually retracted[95]. We expect that a mechanistic understanding of the specific function of *Lmx1-r* in notochord formation will be gained thorough the identification and functional analysis of its target genes, which is currently underway.

Similarly to *Lmx1-r*, the notochord CRM associated with the *Cnot11* locus also requires a Fox binding site; however, the Foxa.a binding data suggest that rather than being activated by Foxa.a, this CRM might be activated by another TF of the Fox family. This scenario is plausible, since at least 28 TFs of the Fox family, in addition to Foxa.a and Foxd, have been identified in *Ciona*, and the expression patterns for some of them are either incomplete or yet to be determined. The identification of notochord CRMs predominantly or exclusively relying on binding sites for Foxa2-related TFs in mouse[33,96,97] suggests that Brachyury-independent subcircuits might be a conserved feature of notochord GRNs across chordates.

The notochord CRMs associated with four of the ten TFs analyzed here (40%) rely solely on Bra/T-box binding sites for their activity. Among them, the *Ror-a* notochord CRM relies on three T-box binding sites, while the remaining ones, *Aff4*, *Etv1* and *Mxd1/Noto7*, rely on individual T-box binding sites. Remarkably, Ci-Bra and Tbx2/3 are, thus far, the only *Ciona* T-box TFs confirmed to be expressed in the developing notochord[73], and we have demonstrated that Tbx2/3 is directly controlled by Ci-Bra and acts as its transcriptional intermediary[35]. Based on these findings, we can envision that the late-onset TF *Aff4* could be controlled indirectly by Ci-Bra through Tbx2/3. In addition to these four notochord CRMs, the *Islet1* CRM also relies on a Bra/T-box binding site for its activity, although a HD binding site is

equally required for the function of this regulatory region. Nevertheless, in embryos ectopically expressing Ci-Bra, the *Islet1* notochord CRM is ectopically expressed as well, similarly to the other CRMs directly targeted by Ci-Bra. This result suggests that the HD activator that is required for the function of this CRM could be either Islet1 itself, thus implying the existence of a positive autoregulatory loop, or a different HD protein, which is either responsive to the ectopic expression of Ci-Bra or is broadly expressed in notochord, CNS and endoderm. This study has also identified a notochord CRM, within the *Cnot11* locus, which requires both a Bra/T-box and a Fox binding site. We have provided the first reports of this class of notochord CRMs synergistically regulated by T-box and Fox proteins[32,58], and suggested that binding sites for these activators might represent evolutionarily conserved building blocks of notochord CRMs that recur in divergent chordates[58]. The recent report of notochord enhancers containing combinations of Bra/T-box and Fox binding sites identified in mouse[98] lends support to this hypothesis.

The remaining two CRMs rely on binding sites for activators of different families. We had gathered the first evidence of a role of AP-1 (activator protein 1) complexes[99] in the regulation of notochord gene expression in a previous survey of notochord CRMs, and we had described a CRM whose activity depends on a combination of AP-1, Fox and HD binding sites[58]. In this study, we found that the *CasZ1* notochord CRM requires a Fox binding site and a TGAC AP-1 half-site (half-AP-1[100]). The *Spalt-like* notochord CRM requires for its activity an E-box. Several TFs of the bHLH family are expressed in the larval CNS of *Ciona*[101], and at least two of them, Mxd1/Noto7[102] and Bhlh-tun1[20,85], are expressed in the developing notochord. The early expression of *Spalt-like* suggests that the activator of its notochord CRM might be Bhlh-tun1, since *Bhlh-tun1* is detected early in notochord precursors[20]. We had previously found that the *Bhlh-tun1* notochord CRM, in turn, is activated by Ci-Bra, Foxa.a, and an early-expressed HD protein[85], which could be either Mnx[87,103] or Lhx3/4/5[20,23,87].

## Multileveled cross-regulatory interactions within the *Ciona* notochord GRN

Even though we have described the Foxa.a-downstream and the Ci-Bra-downstream cascades as seemingly separate circuits of the notochord GRN, multiple lines of evidence suggest that these pillars of notochord development are interconnected at different levels, rather than operating in parallel. In fact, morpholino-oligonucleotide mediated inactivation of Foxa.a leads to the down-regulation of *Ci-Bra*[23], the *Foxa.a* locus is bound by Ci-Bra in early embryos[54]. In addition to this central cross-regulatory interaction, which is conserved across chordates[33,104], our group has recently reported the positive feedback of Xbp1, another Ci-Bra-downstream notochord TF, on Ci-Bra itself, and we have verified the conservation of this subcircuit of the notochord GRN in *Xenopus*[36]. Interestingly, even though *Lmx1-r* had surfaced as one of the potential transcriptional targets of Xbp1[36], the main activator of the *Lmx1-r* notochord CRM is Foxa.a, an additional indication that Ci-Bra would only play a minor indirect role, if any, in the transcriptional regulation of this TF.

Several putative components of vertebrate AP-1 complexes have been identified in the *Ciona* genome[105] and at least one of these genes, *Fos-a*, is expressed in the developing notochord[25]. We had reported evidence that Ci-Bra is required for the notochord expression of *Fos-a*[25] and that Tbx2/3 is responsible for fine-tuning its expression, as *Fos-a* is down-regulated by Tbx2/3 in microarray experiments[35]; together, these findings suggest that Ci-Bra might also modulate the expression of AP-1 complex(es).

## Features of the notochord CRMs associated with *Ciona* TFs

Among the main defining features of *cis*-regulatory regions are their interspecific conservation[106], the accessibility of their genomic contexts[107] and their occupancy by TFs[108]; hence, we have analyzed, retrospectively, the conformity of the notochord CRMs identified in this study to each of these criteria. We found conservation of the *cis*-regulatory regions and their activator binding sites between *C. robusta* and its close relative *C. intestinalis*, while the degree of sequence conservation between *C. robusta* and its distant congener *C. savignyi* dropped to ~50% of the functional binding sites.

Overall, the *cis*-regulatory regions identified in this study display a modular organization, consisting of compact, separable CRMs that are able to function on a heterologous promoter (the *Foxa.a* basal promoter region[56]). With respect to chromatin accessibility, is it noteworthy that the ATAC-Seq datasets used as a reference in this study originated from whole-embryo preparations[52], and the mesenchyme-specific ATAC-Seq dataset[53] likely includes enhancers/genes that are active/expressed in both mesenchyme and notochord. Bearing in mind this important *caveat*, we analyzed the chromatin accessibility of regions overlapping with all the notochord CRMs identified. The chromatin of most notochord CRMs appears accessible throughout most of the developmental stages analyzed, and we did not detect substantial differences in accessibility, even when we compared the chromatin landscapes of the *Fli1/Erg* locus, the TF with the narrowest window of notochord expression, to that of *CasZ1*, which is expressed in notochord cells for the longest amount of time. This might suggest that most of these *cis*-regulatory regions remain readily accessible to their respective activators, keeping the genes that they control primed to be rapidly deployed, as the notochord cells complete morphogenesis and reach terminal differentiation within less than 14 hours. In the case of TFs characterized by narrow windows of expression, the observation that the chromatin accessibility profiles do not match the short pulses of their notochord expression suggests that the activity of these *cis*-regulatory sequences might rely on the availability of their activators, rather than on the accessibility of their genomic regions.

The last criterion that we used to analyze the CRMs, after determining the TF binding sites necessary for their function, was their occupancy by TFs in early embryos, which had been determined through ChIP-on-chip experiments for Foxa.a, Ci-Bra and Foxd[54] and through ChIP-Seq experiments in the case of Foxd[55]. With the exception of the *Mxd1/Noto7* notochord CRM, all Ci-Bra-downstream CRMs were bound by this TF at the early stages analyzed, as were the predicted Fox targets, largely supporting the results of our site-directed mutagenesis experiments.

In the case of *Cnot11*, we cloned and tested all of the regions of accessible chromatin identified by ATAC-Seq experiments, and found that only the region reported here possesses notochord enhancer activity. This result suggests that the presence of multiple enhancer regions active in the same tissue, which has been reported for the *Ci-Bra* and *Hox1* genomic regions[103,109], may not be a feature shared by all *Ciona* TFs. We also noticed that even though *Aff4* and *Cnot11* are both located on chromosome 2, separated by ~53.7 kb, they are associated to independent notochord CRMs that require different transcriptional activators.

## Employing the information gathered from the *Ciona* notochord GRN to reconstruct notochord regulatory circuitries in vertebrates

Mouse orthologs of *Ciona Lmx1*, *Aff4*, *Ror-a*, *CasZ1*, *Islet1*, and *Spalt-like* have been detected in node/notochord cells of developing mice in scRNA-Seq experiments[47]. In humans, *ISLET1* is one of the specific markers of the developing notochord[110].

Despite a large body of evidence in support of the molecular homologies between the notochord of widely different chordates, the information on notochord CRMs associated with TFs in vertebrates

remains fragmentary, derives from different model organisms, and has been mostly focused on the notochord enhancers associated with *Bra/T/Tbxt* and *Foxa2* orthologs. *Cis*-regulatory regions active in notochord cells have been identified in association with mouse *Brachyury/T*[98,111] and, very recently, with human BRACHYURY/T/TBXT[112], with *no tail* (*ntl*), one of the *Brachyury* orthologs in zebrafish[113], and with the promoter region of chick *Brachyury*[114]. A node/notochord enhancer located 15 kb upstream of mouse *Foxa2* has been identified and characterized[115,116]. Germane to these studies, an enhancer region has also been identified for mouse *Islet1*, which in addition to being strongly expressed in notochord and somites[117] is an evolutionarily conserved marker of a population of progenitor cells that give rise to different cardiac structures, including the sinoatrial node of the heart[118]; however, the activity of this mouse *Islet1* enhancer is limited to the sinoatrial node, and does not include the notochord[107]. Similarly, an enhancer located ~500 kb upstream of human *SALL1*, identified by testing human sequences for *cis*-regulatory activity in mouse embryos, only contains the regulatory elements sufficient to recapitulate the expression of this gene in limbs, and is not active in the notochord[119].

We have also analyzed ChIP studies carried out in vertebrates, in an attempt to outline discrepancies and similarities in the modes of regulation of genes encoding notochord TFs across chordates. Studies in differentiating mouse embryonic stem cells indicate that *Foxa2* and *Lmx1b* are among the TFs loci bound by Bra[120]. The occupancy of the *Lmx1b* locus by mouse Bra resembles the occupancy of the *Lmx1-r* locus by Ci-Bra, and this might either indicate that also in mouse the binding of Bra to this enhancer may not necessarily result into a detectable transcriptional output, or that, alternatively, after the duplication of the *Lmx1* gene in vertebrates, at least one of the mouse *Lmx1* orthologs has been incorporated into the Bra-downstream gene battery. As for *Aff* orthologs, the genomic locus of *Aff4* is reportedly bound by Bra in activin-treated mouse embryoid bodies, which suggests that the relationship identified in *Ciona* between these TFs might be conserved across chordates[104]. On the other hand, in mouse embryoid bodies, *CasZ1* is bound by Bra[104], while the *Ciona CasZ1* minimal notochord CRM is devoid of Bra/T-box binding sites and relies upon an AP-1 binding site.

Similarly to what we found in *Ciona*, *ETV1* is reportedly downstream of BRA in a human cell line established from chordomas[121] and is also downstream of Bra in differentiating mouse embryonic stem cells[120], as well as in activin-treated mouse embryoid bodies[104]. The *SPALT-LIKE 1* (*SALL1*) genomic region is bound by BRA in human embryonic stem cells[122,123] and in mouse embryos[104], and the *Ciona Spalt-like* notochord CRM is bound by Ci-Bra in early embryos[54], even though its main activator is a TF of the bHLH family. Finally, *ISLET1* is a target of BRA in humans[122,123], and in mouse and chick motor neurons Islet1 forms a complex with another HD protein, Lhx3, and in this form is capable of amplifying its own expression[124]; our finding that the *Ciona Islet1* notochord CRM requires both a Bra/T-box binding site and a possibly autoregulatory HD binding site suggests that both the dependency upon Bra and the autoregulatory capability of Islet1 might be present in *Ciona* as well.

In conclusion, this study has elucidated the structure and organization of ten *cis*-regulatory nodes of the *Ciona* notochord GRN, has identified the TFs controlling them through a rigorous mutational analysis of their respective activator binding sites, and has identified sequence/function conservation and drifting of *cis*-regulatory regions between distantly related, fast-evolving chordate species. These results provide a high-resolution view of the fast-deploying GRN that orchestrates notochord morphogenesis in a 'simple' yet informative chordate. The knowledge of *cis*-regulatory mechanisms acquired in *Ciona* provides a foundation for investigations of the roles of these evolutionarily conserved TFs in the formation of notochord and *nuclei pulposi* in vertebrates.

## Methods

### *Ciona robusta* embryo cultures

Adult *Ciona robusta* (formerly *Ciona intestinalis* type A[51]) were purchased from M-REP (Carlsbad, CA, USA) and kept at 19 °C in an aquarium with refrigerated recirculating artificial seawater. After in vitro fertilization and dechorionation, zygotes were electroporated with 50–100 μg of each plasmid, cultured at 16-22 °C for ~4–18 h, fixed and stained essentially as described[56].

### Whole-mount in situ hybridization (WMISH)

*Ciona* embryos fixed at stages ranging from 112-cell to late tailbud were hybridized and stained essentially as described[25,35]. Gene-specific antisense RNA probes were synthesized in vitro using as templates ESTs from the *Ciona* Gene Collection release 1[125] and/or the *Ciona* Unigene cDNA collection[126] (Table 1).

### Plasmids construction

Genomic regions from the loci of the notochord TFs of interest were PCR-amplified using standard protocols. After spin-column purification, the PCR-amplified regions were cloned in the pFBΔSP6 plasmid, which contains the *Foxa.a* basal promoter and the *LacZ* reporter gene, and is sporadically active only in a few cells of the mesenchyme and tail muscle[56]. The 163 plasmids containing truncations and site-directed mutations of the notochord enhancer regions were generated by PCR amplification; all oligonucleotide sequences are included in Supplementary Table 4 (*C. robusta*) and Supplementary Table 5 (*C. savignyi*).

To generate the *Sna > Foxa.a::GFP* misexpression construct, the entire open reading frame of *Cr-Foxa.a* was amplified from the Gateway full-ORF cDNA library (clone 93C14)[126] with the primers:

Foxa.a-1761-Fwd-NcoI 5'-tcgccatggATGATGTTGTCGTCTCCACCGTCAAAGTAC-3' and Foxa.a-1761-Rev-SpeI 5'-gtcactagtGCTTGCTGGTACGCACCCTGGGTAGTATGC-3'.

The resulting PCR product was digested with *NcoI/SpeI* and cloned into the *Ci-Bra > En::Tbx2/3^DBD::GFP* plasmid[35], which had been digested with *NcoI/SpeI* to replace the *En::Tbx2/3^DBD* fragment, to create the plasmid *Ci-Bra > Foxa.a::GFP*. The *Foxa.a::GFP* fragment was then excised using *XhoI* and *NcoI*-HF (New England Biolabs, Ipswich, MA, USA) and ligated downstream of the 737-bp *Cr-Snail* muscle enhancer[68] in a vector derived from the *Sna>Foxa.a* misexpression construct[32].

To drive robust and specific expression of Cas9 in the *Ciona* notochord, the *Cas9* coding region was cloned downstream of a truncated version (782-bp long) of the *Ci-Bra* promoter[28], which was PCR-amplified using the following primers:

Bra-782-fwd AscI: 5'-ggcgcgccTGCGTCATTGAGGTTTTGTC-3'

Bra-782-rev NotI: 5'-ggcggccgcCACACTCGGGTGCAAGTTTA-3' and ligated into the *Eef1a -1955/-1 > Cas9* vector[127], which had been digested with *AscI* and *NotI* to excise the *Eef1a* promoter region.

### CRISPR/Cas9 sgRNA design and validation

Single-chain guide RNAs (sgRNAs) targeting exon 2 of the *Lmx1-r* coding region were designed using the CRISPOR software (http://crispor.tefor.net/)[128]. The sgRNAs used in this study were:

Lmxl.ex2.62: 5'-GACGGGGATTTCAGCCACTG (G + N19)

Lmxl.ex2.108: 5'-GAACCGGCGCATAAGACCGG (G + N19)

Control[127]: 5'- GCTTTGCTACGATCTACATT (G + N19)

sgRNA cassettes were custom synthesized and cloned by Twist Bioscience (South San Francisco, CA, USA) into the expression vector *U6>sgRNA (F + E)* backbone[127]. To measure mutagenesis efficacy by Next-Generation Sequencing (NGS), 75 μg of each sgRNA expression plasmid were co-electroporated with 25 μg of *Eef1a -1955/-1 > Cas9::Geminin^N-ter*[129]; after electroporation, embryos were reared in artificial seawater until hatching and collected for genomic DNA extraction. Genomic DNA was isolated with the QiaAMP Micro extraction kit (QIAGEN, Germantown, MD, USA) according to the

manufacturer's instructions, and the targeted region, *Lmx1-r* exon 2, was amplified by PCR using AccuPrime *Pfx* (ThermoFisher, Waltham, MA, USA) and the following primers:

*Lmx1-r* exon 2 Forward: 5'-TTTACTGCCGGTTTCATACGT-3' and
*Lmx1-r* exon 2 Reverse: 5'-TCAAGTCACATATAGCAACGTG -3'.

The resulting PCR products were purified with a QiaQuick PCR Purification kit (QIAGEN, Germantown, MD, USA) and sequenced using commercial Illumina-based NGS Amplicon sequencing (Amplicon-EZ by Genewiz; Azenta Life Sciences, MA, USA). Efficiency was determined by comparing the indel percentage to a control sample electroporated with the *Cas9* vector and *U6 > Control sgRNA*[127], as previously described[130]. Diagrams and plots of sgRNA design and efficiency are shown in Supplementary Fig. 1.

### Immunohistochemistry, microscopy, and image analysis

In vitro fertilized and dechorionated *C. robusta* embryos were cultured in filtered artificial seawater at 21 °C, collected and fixed with 4% paraformaldehyde in phosphate buffered saline (PBS) pH 7.4 for 50 min. at RT. Embryos were washed two times in washing buffer composed of PBS (1x) and Triton X-100 (0.1%), permeabilized with Triton X-100 (0.25%), Tween-20 (0.1%) in PBS for 20 min., followed by two washes in washing buffer, and blocked with bovine serum albumin (BSA, 1%) in washing buffer for 1 h. at 4 °C. Embryos were incubated overnight at 4 °C in the presence of our in-house 1:250 rabbit Ci-Bra[80] and 1:500 mouse beta-galactosidase (cat. number: A-11132; Promega, Madison, WI) antibodies. After that, embryos were washed three times in washing buffer, followed by an incubation in blocking buffer (1% BSA in washing buffer) for 1 h. at 4 °C, then incubated overnight at 4 °C in the presence of goat anti-rabbit Alexa Fluor 488 (cat. number: A27034; Invitrogen, Waltham, MA) and goat anti-mouse Alexa Fluor 555 (cat. number: A21422; Invitrogen, Waltham, MA) fluorescent secondary antibodies, both diluted 1:1000, in blocking buffer. To detect expression of Cas9 in notochord cells, we used the Cas9 antibody 4G10 (cat. number: C15200216-10; Diagenode, Denville, NJ), diluted 1:500.

After washing, embryos were stained with 300 μM 4',6-diamidino-2-phenylindole (DAPI, ThermoFisher, Waltham, MA) in washing buffer for 15 min at room temperature, mounted on glass microscope slides using ProLong™ Glass Antifade Mountant (Invitrogen, Waltham, MA) and imaged using a Leica DMi8 SP8 confocal microscope. Fluorescence confocal images were processed using the Fiji Open Source Software (Version: 2.14.0/1.54 f).

### Statistics and reproducibility

Whole-mount in situ hybridization (WMISH) experiments shown in Fig. 1A (*Lmx1-r*), Fig. 3A (*Aff4*) and Fig. 5S (*Spalt-like*) have been carried out repeatedly prior to their original publication (José-Edwards et al.[25]). The experiment in Fig. 3M (*Mxd1*) has been carried out twice by two different researchers, and the experiment in Fig. 5M (*Islet1*) has been carried out independently in the laboratories of each group of authors. All other WMISH experiments (Figs. 3G, S and 5A, G, Y) have been carried out once because the results were consistent with previously published in situ data from other groups[20,125,131–135].

Each transgenic experiment was carried out at least 3 times in the same conditions, using animals collected on different dates from the same geographic location.

The experiments shown in Fig. 4 were carried out once by immunofluorescence and at least twice using the X-gal staining method, on different batches of embryos. The in-house Ci-Bra antibody has been extensively validated for use in immunofluorescence and ChIP assays, before and after its publication in peer-reviewed scientific journals (Aihara et al.[80]; Katikala et al.[57]).

Graphs and statistical analysis (Student's *t* tests) were generated using GraphPad Prism Version 10.0.3 for Mac OS X (GraphPad Software, San Diego, CA). No statistical method was used to predetermine sample size. No data were excluded from the analyses. The experiments were not randomized, and the investigators were not blinded to allocation during experiments and outcome assessment.

### Reporting summary

Further information on research design is available in the Nature Portfolio Reporting Summary linked to this article.

## Data availability

All data generated during this study are included in this published article and its supplementary information files. Source data are provided with this paper. ATAC-Seq data from *C. robusta* embryos at 64-cell, 112-cell, late gastrula and mid-neurula[52] used in this study are available in the Aniseed and Ghost databases; raw reads were deposited in the NCBI Sequence Read Archive (BioProjects PRJNA474750 and PRJNA474983). ATAC-Seq data from *C. robusta* embryos at the mid-tailbud I stage[53] are available in the Aniseed and Ghost databases and are deposited in the GEO database under accession number GSE126691. ChIP-on-chip data[54] used in this study are available in the Aniseed and Ghost databases and are deposited in the GEO database with the following accession numbers: GSM300069/GSM300071 and GSM300471/GSM300475 (Ci-Bra); GSM441213/GSM441214 and GSM441215 (Foxa.a); GSM441225/GSM441226 and GSM441227 (FoxD); GSM271902/GSM271903 and GSM271904/GSM271905 (ZicL).

ChIP-Seq data (FoxD)[55] used in this study are available in the Aniseed and Ghost databases and are deposited in the NCBI SRA database under accession number DRA005285.

Data on the H3K4me3 promoter mark, obtained by ChIP-Seq data from whole early-gastrula embryos[132], are available in the Aniseed and Ghost databases and are deposited in the NCBI BioProject database, under accession number PRJNA475019.

Original imaging data are available from the authors upon request. Source data are provided with this paper.

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

## Acknowledgements

We thank Ms. Aakarsha Pandey for the *Bra > Cas9* plasmid, and Ms. Diana Kim and Ms. Inthisar Kunnath for their excellent technical help. Thanks to Drs. Yutaka Satou (Kyoto University) and Patrick Lemaire (CRBM-CNRS, University of Montpellier) for permission to use screenshots of the Ghost and Aniseed websites, respectively. We are indebted to Drs. Nigel Bunnet and Dane D. Jensen (Dept. of Molecular Pathobiology, NYU College of Dentistry) for the use of the confocal microscope and guidance on image acquisition. Research reported in this publication was supported by the Eunice Kennedy Shriver National Institute of Child Health and Human Development of the National Institutes of Health, under awards number R03HD098395 and R03HD107314, by a MEGA seed grant from the NYU Office of the Vice Provost for Research and by the Department of Molecular Pathobiology Accelerator Award B01 2020 to A.D.G. This research was also supported by NIH award R01HD104825 to A.S. L.J.N.-P. was supported in part by NIH training grant T32HD007520. D.S.J.-E. was supported in part by NIH training grant T32GM008539.

## Author contributions

A.D.G. designed research; L.J.N.-P., Y.W., S.P., D.S.J.-E. and A.S. performed experiments; L.J.N.-P., Y.W., S.P., A.S. and A.D.G. analyzed data; L.J.N.-P. and A.D.G. prepared figures and tables and wrote the manuscript.

## Competing interests

The authors declare no competing interests.
