## [Peer Review File · Nature Communications]

Cis-regulatory interfaces reveal the molecular mechanisms underlying the notochord gene regulatory network of *Ciona*REVIEWER COMMENTS

Reviewer #1 (Remarks to the Author):

This study addresses the gene regulatory network of notochord formation in ascidian embryos, which may provide insights into how this axial structure appeared during animal evolution. The authors identified and analyzed cis-regulatory modules (CRMs) associated with 10 transcription factors that are expressed in developing notochord both in *Ciona* and vertebrates. Mutation analyses of putative TF binding sites revealed that Fox, Tbx, bHLH, Homeobox and AP-1 transcription factors are candidates driving the transcriptional activation. Even though the CRMs contain different combinations of these binding sites, only one type of binding site was found to be required for each CRM. Available CHIP-seq datasets indicate that the Fox and Tbx-binding sites could be bound by Foxa.a and Brachyury, the factors known to be involved in notochord formation both in *Ciona* and vertebrates. The results of the study are well summarized in Figure 5, highlighting basically two main parallel inputs, Foxa.a and Brachyury, activating target genes in developing notochord cells. This has been partly shown in Reeves et al (2021, Development), in which tissue-specific ectopic expressions of Brachyury and its knock-out, combined with transcriptome analyses, revealed that this TF could control only a subset of notochord genes and that Foxa.a and Mnx were likely-candidates acting in parallel to activate additional sets of notochord genes. The current study supports this conclusion by painstakingly analyzing ten notochord-CRMs. However, while the experiments seem to be carefully conducted, the study mainly provides supportive evidence for what was previously reported. Furthermore, it lacks precise functional studies such as notochord-specific knock-out of *Foxa.a*, *Tbx2/3* and candidate TFs for Hox, bHLH and AP-1 binding sites.

As some of them are discussed in the manuscript, there are several questions that could be of more general interest in the field: how is the temporal order of *Brachyury*-dependent target genes controlled? (I am aware that the DiGregorio lab published an article on this aspect but the underlying mechanism remains unknown); why are Foxa.a-dependent genes not expressed in the other *Foxa.a*-positive axial structures?; how are genes encoding notochord structural proteins (e.g., sheath, vacuoles) activated?-is there a specific sub-circuit for each subcellular structure?

I list below a few additional comments concerning the experiments reported in the manuscript:

- 1) In terms of the “trans-activation assays” in Figure 3, I wonder whether Bra could activate CRMs when expressed in ectoderm lineages. If the exogenous Bra is tagged, it will be much easier to appreciate the effect of Bra on these modules. Similarly, it will be helpful if the exogenous Foxa.a is also tagged to detect it together with LacZ in Figure 1M-P.
- 2) In terms of chromatin accessibility, it is better to generate a notochord-specific profile using FACS-isolated developing notochord cells.
- 3) I recommend the authors to address if the *C. savignyi* counterparts of CRMs drive notochord expression in *Ciona robusta*.
- 4) The “functional” study shown in Figure 1Q,R is not convincing and should be replaced by a CRISPR/Cas9-mediated gene knockout or morpholino-mediated gene knockdown followed by a more thorough analysis of the phenotype.
- 5) When Bra2 in the *Mxd1* CRM is mutated, the reporter gene becomes activated in midline ectoderm cells (Fig S4C). Any comment on this?
- 6) The authors describe the effect of *Foxa.a>Bra* on the *Etv1* CRM as “a nearly complete overlap...” (page 15, line 21). It is hard to appreciate this statement from the images

presented in Figure 3J-L.

7) I cannot find the data showing that the Fox binding site in the *Cnot11*-CRM is dispensable (page 18, line 1).

8) The authors discuss a potential involvement of other Fox family TFs than Foxa.a in regulation of *Fli1/Erg* (page 23, line 9). They should address whether some of them are expressed in notochord cells.

Reviewer #2 (Remarks to the Author):

This manuscript “Cis-regulatory interfaces reveal the molecular mechanisms underlying the notochord gene regulatory network of *Ciona*” by Negrón-Piñeiro et al. expands on our understanding of the regulatory connections used to build the gene regulatory network of the notochord. The authors laid the foundations to get important informations on the intricate mechanisms controlling an evolutionary hallmark of the chordate phylum by collecting and assembling a huge amount of data, including detailed cis-regulatory analyses of ten TFs expressed in the notochord with those of trans-activation assays.

The output is undoubtedly of great interest, the paper is well written and conclusive by illustrating how two evolutionarily conserved TFs, Brachyury and Foxa2, coordinate the deployment of the notochord GRN. Overall, a publication is recommended.

Few points need to be addressed to better clarify the text and the connection between figures and text:

The authors should have numbered the text figures and the supplementary figures to easily find them when reading the text.

Page 11 “The remaining two Bra/T-box binding sites were mutagenized within a 95-bp truncated region that is still able to yield reliable notochord expression, and only one of them was found to be necessary for the activity of the 145-bp CRM”. What about the 95-bp truncated region? It appears only in this sentence and not elsewhere.

Page 14 “This region was further narrowed to a 562-bp CRM, which is enriched in Bra/T-box binding sites, four of which, however, are included in a 129-bp region active only in endodermal cells (Fig. 2V)”. The authors should add at the end (Fig. 2V and yellow box in Fig. 2T)

Page 16 “Instead, Ci-Bra is required for the notochord expression of *Aff4* (Jose'-Edwards et al., 2011) and for the activity of its notochord CRM, and is sufficient to evoke the ectopic expression of this notochord CRM and of those associated with *Mxd1* and *Etv1*, as well as of other CRMs (Fig. S8; see below)”. Why ROR is not included in the genes directly regulated by CiBra?

Page 16 The authors should add a brief introduction on AP-1 TFs which, based on the authors data, seems to be involved in the activation of *CasZ1*

Pages 18-19 The authors should move (Fig. 4R) at the end of the sentence “this region might be contacted by Fox proteins during early embryogenesis”

Figure legend 2 “Identification of notochord CRMs within the genomic loci of the genes in (A,G,M,S,Y)” Eliminate Y since the panel ends with S

Figure legend 2 “ asterisk in (T) refers to a region that we had previously analyzed in Katikala et al., 2013.” The asterisk is in N non in T

Reviewer #3 (Remarks to the Author):

The authors focused on the role of two evolutionarily conserved key transcription factors, Brachyury and Foxa2, in gene regulation of another ten evolutionarily conserved transcription factors involved in the notochord development of the ascidian *Ciona robusta*. The authors experimentally identified notochord-specific cis-regulatory modules (CRMs) and analyzed function of these CRMs in detail. As the results, the authors established higher resolution gene regulatory networks (GRNs) for development of the notochord, the characteristic of chordates.

Molecular mechanisms of notochord development in ascidians is an important topic that has attracted the attention of many researchers and has been the subject of numerous studies. Among them, the present study is an unprecedented level of detailed functional analysis of the cis-regulatory regions of each of the 10 genes encoding transcription factors, and the experiments and data are such that a general paper could be written on each individual gene. The authors further also took advantage of deep accumulation of data concerning chromatin accessibility and occupation of transcription factors on the genomic DNA regions in *Ciona robusta* to interpret their experimental data and construct gene regulatory networks. This work surely deepens the understanding of GRNs. On the other hand, the conclusions reached by the authors in this study are an extension of previous findings and do not force a major change in our understanding of the mechanisms of notochord development in ascidians. Because the orthologues of the transcription factors they analyzed are all expressed in the notochord (and/or its developmental derivative nucleus pulposus) of vertebrates, their results should provide important information for the study of GRNs for notochord development in vertebrates.

This study is an integration of laborious and high quality experiments. The manuscript is well written, the methodology is sound, and their transgenic data are solid and beautiful. The data provided by the authors supports their conclusions and claims.

Point-by-point response to the Reviewers' comments, reproduced verbatim

Reviewer #1

This study addresses the gene regulatory network of notochord formation in ascidian embryos, which may provide insights into how this axial structure appeared during animal evolution. The authors identified and analyzed cis-regulatory modules (CRMs) associated with 10 transcription factors that are expressed in developing notochord both in *Ciona* and vertebrates. Mutation analyses of putative TF binding sites revealed that Fox, Tbx, bHLH, Homeobox and AP-1 transcription factors are candidates driving the transcriptional activation.

Even though the CRMs contain different combinations of these binding sites, only one type of binding site was found to be required for each CRM.

Response: This last statement is incorrect. In Fig. 4 we show that three CRMs require for their activity two different types of binding sites, *i.e.*, the *CasZ1* CRM requires Fox and AP-1, *Cnot11* requires Fox and Bra/T-box, and *Islet1* requires Bra/T-box and homeodomain binding sites.

Available ChIP-seq datasets indicate that the Fox and Tbx-binding sites could be bound by Foxa.a and Brachyury, the factors known to be involved in notochord formation both in *Ciona* and vertebrates. The results of the study are well summarized in Figure 5, highlighting basically two main parallel inputs, Foxa.a and Brachyury, activating target genes in developing notochord cells. This has been partly shown in Reeves et al (2021, Development), in which tissue-specific ectopic expressions of Brachyury and its knock-out, combined with transcriptome analyses, revealed that this TF could control only a subset of notochord genes and that Foxa.a and Mnx were likely-candidates acting in parallel to activate additional sets of notochord genes.

Response: To provide some context to these statements, we need to point out that our group first identified *Lmx1-r* and reported its expression in notochord in 2011 (Jose'-Edwards et al., 2011), and it was at that time that we first set forth the original hypothesis that this TF might act independently of Brachyury and thus be part of a Bra-independent branch of the notochord GRN. With this current work, we have not only verified our hypothesis, but we have determined that the main activator of the *Lmx1-r* notochord CRM is Foxa.a. In addition, in this extensively revised manuscript we provide the first report of the notochord phenotype caused by the loss of function of this gene.

Of note, some of the conclusions of the study published by Reeves et al., 2021 had been already outlined by the foundational work of Imai et al., 2006, which had uncovered the regulatory interactions between Foxa.a and Brachyury through a rigorous morpholino-mediated inactivation of most of the *Ciona* transcription factors, and by the work of Kubo et al., 2010, who had identified numerous regulatory interactions through their ChIP-on-chip data.

However, neither Imai et al., 2006 nor Reeves et al., 2021 nor Kubo et al., 2010 have analyzed these regulatory interactions at the enhancer level. Studying regulatory interactions at the enhancer level provides a view of the GRN from a unique angle, and

has shed light on synergistic interactions between activators, on the structure and function of notochord enhancers, on the sequences of functional binding sites required for notochord gene expression, on the interpretation of ATAC-Seq profiles, ChIP results and on interspecific and evolutionary conservation of *cis*-regulatory information.

The current study supports this conclusion by painstakingly analyzing ten notochord-CRMs. However, while the experiments seem to be carefully conducted, the study mainly provides supportive evidence for what was previously reported.

Response: We respectfully but firmly disagree with this belittling judgment of the present study. In addition to the overall significance of this study, which we have outlined above, we provide the first report of the notochord phenotype caused by the loss of function of *Lmx1-r*, the direct evidence of the requirement of a *Foxa.a* binding site for its notochord activity, and the sufficiency of *Foxa.a* binding site for the ectopic expression of *Lmx-1r* in muscle cells. We report the cooperative function of Bra/T-box sites in the activation of the *Ror-a* notochord CRM, the synergistic activation of the *CasZ1*, *Cnot11* and *Islet1* by different TFs, and provide the first evidence of the requirement of both a Bra/T-box and a (possibly autoregulatory) homeodomain binding site within the *Islet1* CRM.

We provide a study that complements, and at times challenges, existing chromatin accessibility and TF occupancy data, and a thorough analysis of the interspecific conservation of all these notochord enhancers between divergent *Ciona* species.

To the best of our knowledge, none of these findings were previously reported, and none of the previous reports has achieved this level of resolution.

Furthermore, it lacks precise functional studies such as notochord-specific knock-out of *Foxa.a*, *Tbx2/3* and candidate TFs for Hox, bHLH and AP-1 binding sites.

Response: In this revised manuscript we provide a precise functional study, and the notochord-specific knock-out of *Lmx1-r*, which has been obtained through CRISPR-Cas9-mediated genome editing. All the additional knock-outs that Reviewer #1 finds are lacking from this study would certainly provide insightful results, but they do not seem indispensable to complement the nearly exorbitant amount of experiments that have been carried out for this manuscript, and seem a bit beyond the scope of this study.

As some of them are discussed in the manuscript, there are several questions that could be of more general interest in the field: how is the temporal order of Brachyury-dependent target genes controlled? (I am aware that the DiGregorio [*sic*] lab published an article on this aspect but the underlying mechanism remains unknown);

Response: As Reviewer #1 correctly points out, we have set forth a mechanistic hypothesis in Katikala et al., (*PLOS Biology*, 2013); we are currently completing this separate line of research. The focus of this revised manuscript remains to shed light on the structure of the notochord GRN using the information gathered from the detailed analysis of notochord CRMs associated with TFs.

...why are Foxa.a-dependent genes not expressed in the other Foxa.a-positive axial structures?;

Response: As we had already hypothesized in the Discussion section, it is possible that the expression in other Foxa.a-positive axial structures might be counteracted by tissue-specific repressors. In the case of *Lmx1-r*, this hypothesis seems to be supported by the ectopic endodermal staining that is detected in transgenic embryos carrying truncated versions of the *Lmx1-r* notochord CRM, a pattern that could be attributed to the removal of binding sites for transcriptional repressors.

...how are genes encoding notochord structural proteins (e.g., sheath, vacuoles) activated?-is there a specific sub-circuit for each subcellular structure?

Response: This is a very interesting question. It seems that Reviewer #1 is referring to extracellular structures rather than subcellular, such as the vacuoles (more correctly ‘extracellular pockets’; Denker and Jiang, 2012) that form the lumen of the notochord by coalescence, and the notochord sheath. Our previous studies on the enhancers of structural notochord genes have uncovered how Ci-Bra controls, either directly or through *Tbx2/3*, the notochord CRMs of tropomyosin-like (Di Gregorio and Levine, 1999), leprecan/P3H1 (Dunn and Di Gregorio, 2009), fibrillar collagen, thrombospondin 3, ezrin-radixin-moesin and others (Katikala et al., 2013; Thompson and Di Gregorio, 2015, etc.). Our studies of the target genes of *Ciona Xbp1* (Wu et al., *Elife* 2022) suggest that the Bra-Xbp1 cross-regulatory sub-circuit controls the expression of an evolutionarily conserved core of genes required for unfolded protein response, and also the expression of some genes whose products are ECM components and might be involved in the formation of the notochord sheath.

I list below a few additional comments concerning the experiments reported in the manuscript:

1) In terms of the “trans-activation assays” in Figure 3, I wonder whether Bra could activate CRMs when expressed in ectoderm lineages.

Response: The effects of the expression of Ci-Bra in pan-neural and muscle lineages have been reported by Reeves et al. (2021). Even though in these studies Ci-Bra has not been specifically expressed in fate-restricted ectoderm/epidermal cells, and this research was not carried out at the enhancer level, some interesting conclusions relevant to this question have been drawn regarding the ability of Ci-Bra to induce ectopic expression of notochord genes, either alone or in combination with Foxa.a and *Mnx* (Reeves et al., 2021).

If the exogenous Bra is tagged, it will be much easier to appreciate the effect of Bra on these modules.

Response: We have shown that our antibody successfully detects both the endogenous (Fig. 3A,G) and the exogenous (Fig. 3D,J,M) Bra protein. We feel that even if successful,

tagging the exogenous Bra in Fig. 3 would not improve much the detection of the effect of Bra on these notochord CRMs, which is the main point that we are trying to make.

Similarly, it will be helpful if the exogenous Foxa.a is also tagged to detect it together with LacZ in Figure 1M-P.

Response: In this case, we have welcomed the Reviewer's suggestion because even though we had successfully used the Sna>Foxa.a construct in Passamaneck et al., 2009, we did not have a reliable read-out of the misexpression of Foxa.a in muscle cells. Therefore, we have tagged Foxa.a with GFP, and we now show the results of the co-electroporation of the tagged protein and both the WT and mutant Fox Lmx-1r notochord CRM in the revised Fig. 1.

2) In terms of chromatin accessibility, it is better to generate a notochord-specific profile using FACS-isolated developing notochord cells.

Response: We completely agree with Reviewer #1 and we are in the process of generating this resource for future studies.

3) I recommend the authors to address if the *C. savignyi* counterparts of CRMs drive notochord expression in *Ciona robusta*.

Response: To address this important point, we have cloned all the *C. savignyi* counterparts of the *C. robusta* CRMs and we have tested them in triplicate in *C. robusta* embryos. We have added photos of a representative embryo to each supplemental figure, and we have commented on the results of these experiments and their possible explanations.

4) The “functional” study shown in Figure 1Q,R is not convincing and should be replaced by a CRISPR/Cas9-mediated gene knockout or morpholino-mediated gene knockdown followed by a more thorough analysis of the phenotype.

Response: We have followed this recommendation and we have carried out CRISPR/Cas9-mediated gene knockout of *Lmx1-r*. The new results are shown in revised Fig. 1 and the sgRNAs are mapped to the *Lmx1-r* coding region in Fig. S1. The notochord phenotypes have been carefully documented and quantified in triplicate with appropriate controls.

5) When Bra2 in the Mxd1 CRM is mutated, the reporter gene becomes activated in midline ectoderm cells (Fig S4C). Any comment on this?

Response: The sequence of this mutated site (ACACTGTAATAA, shown here in reverse and complement orientation for clarity) is similar to a Sox8 binding site (aCACTGNAATgtt; Jolma et al., 2013). In *Ciona* there are two TF of the Sox family, SoxB1 (SOX14; SOX2; SOX21; gene model KH.C1.99) and SoxC (SOX11, SOX12, SOX4; KH.C7.523) that are expressed in epidermal precursors and in the midline

epidermis, respectively; we might have inadvertently created a binding site for one of these TFs in the 216-bp *Mxd1* Bra Mt-2 construct (currently renamed Bra2Mt for clarity). We have inserted this observation in the main text.

6) The authors describe the effect of *Foxa.a*>Bra on the *Etv1* CRM as “a nearly complete overlap....” (page 15, line 21). It is hard to appreciate this statement from the images presented in Figure 3J-L.

Response: We agree that this statement could have seemed misleading. To remediate, we have added an inset in Fig. 3L that shows a close-up of the ventral region of the embryo obtained by merging only the red (beta-galactosidase antibody) and the green (*Ciona* Brachyury-specific antibody) channels. This image shows that numerous cells that express Ci-Bra (Fig. 3J) also contain the 159-bp *Etv1*>*LacZ* transgene (red in Fig. 3K). The incomplete overlap is likely due to differences in efficiency (“strength”) between the drivers, *i.e.*, the 3.5-kb *Foxa.a* promoter region, which drives Ci-Bra expression, and that of the narrowed-down *Etv1*>*LacZ* transgene. However, the ectopic staining can be appreciated in its entirety in Fig. S7E, where the read-out is the beta-galactosidase enzymatic activity, instead of the levels of beta-galactosidase as a protein. We have clarified these points in the revised text.

7) I cannot find the data showing that the Fox binding site in the *Cnot11*-CRM is dispensable (page 18, line 1).

Response: We are indebted to this Reviewer for pointing out this discrepancy. We had based this conclusion on the results of a couple of truncations of this notochord CRM. Prompted by the Reviewer’s request, we revisited the results of the truncation/mutation analysis and after careful consideration we decided to introduce the Bra/T-box and AP-1 mutations in a longer enhancer region that provides more reliable notochord staining (158-bp in revised Fig. 4I,J,K); within this context, we mutagenized the Fox site as well, and found that it is also required for notochord activity, in addition to the Bra/T-box site. On the basis of these recent results, we have reached the conclusion that the main activators of *Cnot11* are Brachyury and *Foxa.a*. We have thoroughly revised the text and figures pertaining to this CRM.

8) The authors discuss a potential involvement of other Fox family TFs than *Foxa.a* in regulation of *Fli1*/*Erg* (page 23, line 9). They should address whether some of them are expressed in notochord cells.

Response: At least 19 TFs of the Fox family, in addition to *Foxa.a* and *FoxD*, have been annotated in the *Ciona* genome. To date, none of them is reported as being unequivocally expressed in notochord, however, in some cases the available *in situ* patterns are either incomplete or missing altogether. We are currently pursuing this line of investigation. We have expanded on this point, which was already present in the Discussion, in this revised manuscript.

Reviewer #2

This manuscript “Cis-regulatory interfaces reveal the molecular mechanisms underlying the notochord gene regulatory network of *Ciona*” by Negrón-Piñero et al. expands on our understanding of the regulatory connections used to build the gene regulatory network of the notochord. The authors laid the foundations to get important informations on the intricate mechanisms controlling an evolutionary hallmark of the chordate phylum by collecting and assembling a huge amount of data, including detailed cis-regulatory analyses of ten TFs expressed in the notochord with those of trans-activation assays. The output is undoubtedly [*sic*] of great interest, the paper is well written and conclusive by illustrating how two evolutionarily conserved TFs, Brachyury and Foxa2, coordinate the deployment of the notochord GRN. Overall, a publication is recommended. Few points need to be addressed to better clarify the text and the connection between figures and text:

The authors should have numbered the text figures and the supplementary figures to easily find them when reading the text.

Page 11 “The remaining two Bra/T-box binding sites were mutagenized within a 95-bp truncated region that is still able to yield reliable notochord expression, and only one of them was found to be necessary for the activity of the 145-bp CRM”. What about the 95-bp truncated region? It appears only in this sentence and not elsewhere.

Response: We have done our best to improve the citations of the supplementary figures. We agree that the 95-bp region was not described sufficiently. To improve the overall analysis, we have replaced the 95-bp region with a new construct, 102-bp in Fig. 2D, that was built to more efficiently mutagenize two of the T-box/Bra binding sites, and we have improved the presentation of the constructs to the best of our abilities. We have also added to Fig. 2D the depiction of the Fox mutant.

Page 14 “This region was further narrowed to a 562-bp CRM, which is enriched in Bra/T-box binding sites, four of which, however, are included in a 129-bp region active only in endodermal cells (Fig. 2V)”. The authors should add at the end (Fig. 2V and yellow box in Fig. 2T)

Response: According to this suggestion, we have corrected the text as follows: (Yellow bar in Fig. 2T and Fig. 2V).

Page16 “Instead, Ci-Bra is required for the notochord expression of *Aff4* (Jose’-Edwards et al., 2011) and for the activity of its notochord CRM, and is sufficient to evoke the ectopic expression of this notochord CRM and of those associated with *Mxd1* and *Etv1*, as well as of other CRMs (Fig. S8; see below)”. Why ROR is not included in the genes directly regulated by CiBra?

Response: We have co-electroporated the *ROR* enhancer region with the *Foxa.a>Bra* construct, but the results were not clear, since the widespread expression of this CRM in

various tissues rendered indiscernible any signal due to ectopic expression. We have added this clarification to the main text.

Page 16 The authors should add a brief introduction on AP-1 TFs which, based on the authors data, seems to be involved in the activation of CasZ1

Response: A brief discussion of the AP-1 TFs is provided in the Discussion.

Pages 18-19 The authors should move (Fig. 4R) at the end of the sentence “this region might be contacted by Fox proteins during early embryogenesis”

Response: (Fig. 4R) has been moved as requested.

Figure legend 2 “Identification of notochord CRMs within the genomic loci of the genes in (A,G,M,S,Y)” Eliminate Y since the panel ends with S

Response: Reviewer #2 is correct, and we have removed the Y accordingly.

Figure legend 2 “ asterisk in (T) refers to a region that we had previously analyzed in Katikala et al., 2013).” The asterisk is in N non [*sic*] in T.

Response: We are thankful to Reviewer #2 for spotting this mistake, which has now been corrected.

Reviewer #3 (Remarks to the Author):

The authors focused on the role of two evolutionarily conserved key transcription factors, Brachyury and Foxa2, in gene regulation of another ten evolutionarily conserved transcription factors involved in the notochord development of the ascidian *Ciona robusta*. The authors experimentally identified notochord-specific cis-regulatory modules (CRMs) and analyzed function of these CRMs in detail. As the results, the authors established higher resolution gene regulatory networks (GRNs) for development of the notochord, the characteristic of chordates.

Molecular mechanisms of notochord development in ascidians is an important topic that has attracted the attention of many researchers and has been the subject of numerous studies. Among them, the present study is an unprecedented level of detailed functional analysis of the cis-regulatory regions of each of the 10 genes encoding transcription factors, and the experiments and data are such that a general paper could be written on each individual gene. The authors further also took advantage of deep accumulation of data concerning chromatin accessibility and occupation of transcription factors on the genomic DNA regions in *Ciona robusta* to interpret their experimental data and construct gene regulatory networks. This work surely deepen the understanding of GRNs. On the other hand, the conclusions reached by the authors in this study are an extension of previous findings and do not force a major change in our understanding of the

mechanisms of notochord development in ascidians. Because the orthologues of the transcription factors they analyzed are all expressed in the notochord (and/or its developmental derivative nucleus pulposus) of vertebrates, their results should provide important information for the study of GRNs for notochord development in vertebrates.

This study is an integration of laborious and high quality experiments. The manuscript is well written, the methodology is sound, and their transgenic data are solid and beautiful. The data provided by the authors supports their conclusions and claims.

Response: We are thankful to Reviewer #3 for their perspective and their appreciation of the work reported in this manuscript.

REVIEWERS' COMMENTS

Reviewer #1 (Remarks to the Author):

In this revised manuscript, the authors addressed most of my suggestions/comments. I find that the datasets are now clearer and more convincing. This solid study provides experimental evidence that improves our current understanding of the GRN controlling Ciona notochord development, without bringing any major changes.

I have one technical comment for the newly-added CRISPR/Cas9 KO study. I recommend the authors to provide a data showing the efficacy of the sgRNAs (e.g., Sanger sequencing-based peakshift assay).

Reviewer #2 (Remarks to the Author):

The authors have fully and/or well argued the points I raised. My opinion is therefore that this manuscript, NCOMMS-22-44804A, deserves publication in Nature comm.

Point-by-point response to the Reviewers' comments, reproduced verbatim

Reviewer #1

In this revised manuscript, the authors addressed most of my suggestions/comments. I find that the datasets are now clearer and more convincing. This solid study provides experimental evidence that improves our current understanding of the GRN controlling Ciona notochord development, without bringing any major changes.

I have one technical comment for the newly-added CRISPR/Cas9 KO study. I recommend the authors to provide a data showing the efficacy of the sgRNAs (e.g., Sanger sequencing-based peakshift assay).

Response: We have added to the manuscript the efficacy of each sgRNA in both the Methods and the legend to Fig. S1, and we have clarified that these data were obtained through the analysis of indel plots obtained by next-generation sequencing of target amplicons from larvae electroporated with each sgRNA or with a negative control sgRNA.

Reviewer #2

The authors have fully and/or well argued the points I raised. My opinion is therefore that this manuscript, NCOMMS-22-44804A, deserves publication in Nature comm.

Response: We are thankful to Reviewer #2 for their appreciation of the research reported in this manuscript.